# DNA methylation patterns identify subgroups of pancreatic neuroendocrine tumors with clinical association

Vanessa Lakis [1], Rita T. Lawlor[2], Felicity Newell [1], Ann-Marie Patch [1], Andrea Mafficini [2],
Anguraj Sadanandam [3,4], Lambros T. Koufariotis[1], Rebecca L. Johnston [1], Conrad Leonard [1],
Scott Wood [1], Borislav Rusev[2], Vincenzo Corbo[2,5], Claudio Luchini [2,5], Sara Cingarlini[6,7], Luca Landoni [6,8],
Roberto Salvia[6,8], Michele Milella[6,7], David Chang [9,10,11], Peter Bailey[9,12], Nigel B. Jamieson [9,10,11],
Fraser Duthie[9,13], Marie-Claude Gingras [14,15], Donna M. Muzny[14], David A. Wheeler [14],
Richard A. Gibbs[14,16], Massimo Milione[17], APGI*, ARC-Net*, Paolo Pederzoli[18], Jaswinder S. Samra[19],
Anthony J. Gill[19,20], Amber L. Johns[20], John V. Pearson[1], Andrew V. Biankin [9], Sean M. Grimmond [21],
Nicola Waddell [1,22], Katia Nones [1,38✉] & Aldo Scarpa [2,5,6,38✉]

Here we report the DNA methylation profile of 84 sporadic pancreatic neuroendocrine tumors (PanNETs) with associated clinical and genomic information. We identified three subgroups of PanNETs, termed T1, T2 and T3, with distinct patterns of methylation. The T1 subgroup was enriched for functional tumors and *ATRX*, *DAXX* and *MEN1* wild-type genotypes. The T2 subgroup contained tumors with mutations in *ATRX*, *DAXX* and *MEN1* and recurrent patterns of chromosomal losses in half of the genome with no association between regions with recurrent loss and methylation levels. T2 tumors were larger and had lower methylation in the *MGMT* gene body, which showed positive correlation with gene expression. The T3 subgroup harboured mutations in *MEN1* with recurrent loss of chromosome 11, was enriched for grade G1 tumors and showed histological parameters associated with better prognosis. Our results suggest a role for methylation in both driving tumorigenesis and potentially stratifying prognosis in PanNETs.

A full list of author affiliations appears at the end of the paper.

   1

Pancreatic neuroendocrine tumours (PanNETs) are rare epithelial neoplasms derived from neuroendocrine cells of pancreatic islets whose incidence has collectively increased over the last decades[1]. The overall 5-year survival rate for Pan-NET patients is 54%, but varies greatly according to stage at diagnosis. For patients with local disease, the 5-year survival rate is 93%, whereas patients with locally advanced disease it is 77% and for those with distant metastatic disease it is 27% (www.cancer.net/cancer-types/neuroendocrine-tumor-pancreas/statistics; https://www.cancer.org/cancer/pancreatic-neuroendocrine-tumor/detection-diagnosis-staging/survival-rates.html). Recent genome, transcriptome and methylome studies have disclosed PanNETs molecular heterogeneity that reflects their variable clinical course[2–9]. PanNETs are either functional or non-functional with the latter being more prevalent. Functional PanNETs are associated with hormone secretion-associated symptoms that lead to early diagnosis. Non-functional PanNETs, due to the lack of early symptoms, are normally detected at later stages with locally advanced or metastatic disease. Non-functional PanNETs are a more heterogeneous group of tumours with unpredictable and varying degrees of malignancy[1,10]. The best predictor of clinical outcome of PanNETs is the fraction of proliferating neoplastic cells, with high-grade (G3) tumours being more aggressive disease and with molecular alterations that often align them with neuroendocrine carcinomas[1]. Conversely, low-grade (G1) and intermediate-grade (G2) PanNETs have a distinct molecular pathology and lack reliable biomarkers to assist prediction of malignancy and selection of treatment[11].

Previous studies have examined DNA methylation and genomic patterns in an attempt to further subclassify PanNETs. DNA methylation has been shown to play an important role in tumorigenesis in many cancers. Studies evaluating DNA methylation of candidate genes in PanNETs have shown aberrant methylation of specific genes in subsets of tumours[12–14]. Studies have suggested that methylation might contribute to the identification of PanNETs subgroups and these subgroups could potentially be associated with clinical features[7–9]. More recently, studies have suggested that methylation can identify cell of origin for PanNETs[15–17]. A few studies have suggested that recurrent patterns of copy number alterations observed in PanNETs, such as whole chromosome gains or losses including loss of heterozygosity (LOH), have an association with clinical parameters[2,18,19]. Sequencing efforts have identified ATRX, DAXX and MEN1 as the most commonly mutated genes in PanNETs and genes within the mammalian target of rapamycin (mTOR) pathway have been reported at a lower frequency[2,20].

Here we evaluated a large cohort of sporadic PanNETs with whole-genome methylation analysis combined with clinical and other genomic features to provide a more comprehensive picture of molecular features of PanNETs with potential clinical implications.

## Results

**Clinical and mutation information**. A total of 84 clinically sporadic primary PanNETs and 11 normal adjacent pancreata were profiled using the Illumina 450K methylation arrays. The clinical and genomic information from Scarpa et al.[2] (Supplementary Data 1) was used to gain insights into potential PanNETs subgroups identified by methylation patterns. There were 32 females and 52 males, 34 G1, 48 G2, 2 G3 (WHO, 2019) and 11 functional tumours in the cohort. The cohort included 9 tumours harbouring somatic mutations in the ATRX gene, 21 in DAXX and 31 in MEN1. There were also four tumours with MEN1 germline variants (Supplementary Data 1).

**Methylation subgroups**. To identify cancer-specific DNA methylation that could potentially stratify PanNETs, we selected CpG sites located in promoter regions, which were not methylated ($\beta$-value < 0.3) across all 11 normal adjacent pancreata samples. Out of 411,159 CpG sites that passed quality filtering, 161,299 sites were not methylated in the normal pancreata ($\beta$-value < 0.3), 111,113 of which were located in gene promoter regions (TSS1500, TSS200, 5′-untranslated region (5′-UTR), or 1stExon). From the latter 111,113 sites, a total of 3378 CpGs had a $\beta$-value SD > 0.20 of the DNA methylation levels across all tumours. With the probe selection described above, we expected to reduce potential confounding signal from non-tumour cells present in the tumour samples. The methylation levels of these 3378 most variable promoter CpG sites were dichotomized and unsupervised clustering identified three distinct clusters across the tumour samples, suggesting potential subgroups of PanNETs (Supplementary Fig. 1). Similar subgroups were identified when selecting probes across the entire genome (Supplementary Fig. 2)

**Association of methylation subgroups with commonly mutated genes MEN1, DAXX and ATRX, and with clinical features**. The three PanNETs subgroups (termed T1, T2 and T3) identified by methylation patterns (Supplementary Fig. 1) showed potential associations with clinical and genomic features (Fig. 1a). Subgroup T1 represents 26.2% of the cohort and harboured significantly fewer mutations in the three most commonly mutated genes in PanNETs (ATRX, DAXX and MEN1) (p-adj ≤ 0.02, Fig. 1b). This subgroup was enriched for functional tumours (7 out 11 functional tumours, Supplementary Data 1). Subgroup T2 represented 42.9% of the cohort; tumours in this subgroup were enriched for mutations in ATRX and DAXX (p-adj ≤ 0.003, Fig. 1c), were significantly larger than tumours in the other two subgroups (p-adj ≤ 0.0003, Fig. 1d), had longer telomeres (p-adj ≤ 0.00008, Fig. 1e) and presented a high frequency of alternative lengthening of telomeres (p-adj ≤ 0.0001, Fig. 1a). Tumours in the T2 subgroup also had a higher tumour mutation burden (point mutations per Mb) (p-adj ≤ 0.03, Fig. 1f) and harboured 80% of the mutations in mTOR pathway genes (Supplementary Data 1). There was no difference in age of the patients or gender distribution between the subgroups; however, tumour content was slightly lower in tumours in the T1 subgroup than the other two subgroups (p-adj = 0.018, Fig. 1a and Supplementary Data 1). Tumours in subgroup T3 harboured mutations in MEN1 (contained 30.9% of the cohort), had a higher proportion of G1 tumours (p-adj = 0.001, Fig. 1a) and a lower frequency of extra-pancreatic spread, perineural and vascular invasion (Fig. 1g–i), suggesting a subgroup of tumours with better prognosis.

**Association of methylation subgroups with somatic copy number changes**. Previous studies have reported subgroups of PanNETs associated with copy number changes[2,5]. To investigate how copy number alterations relate to the methylation subgroups (T1–T3), we compared copy number data generated by Scarpa et al.[2] with the methylation data (Fig. 2). Subgroup T1 had heterogeneous profiles of copy number and were mainly wild-type for ATRX/DAXX/MEN1. Subgroup T2 was enriched for tumours with recurrent LOH in chromosomes 1, 2, 3, 6, 8, 10, 11, 15, 16, 21 and 22. Tumours in the T3 subgroup had mainly diploid genomes with recurrent loss of chromosome 11. We next investigated whether clustering using tumour methylation could be influenced by the different profile of copy number observed between the groups, more precisely if loss of copy number would result in loss of methylation. We evaluated the methylation levels

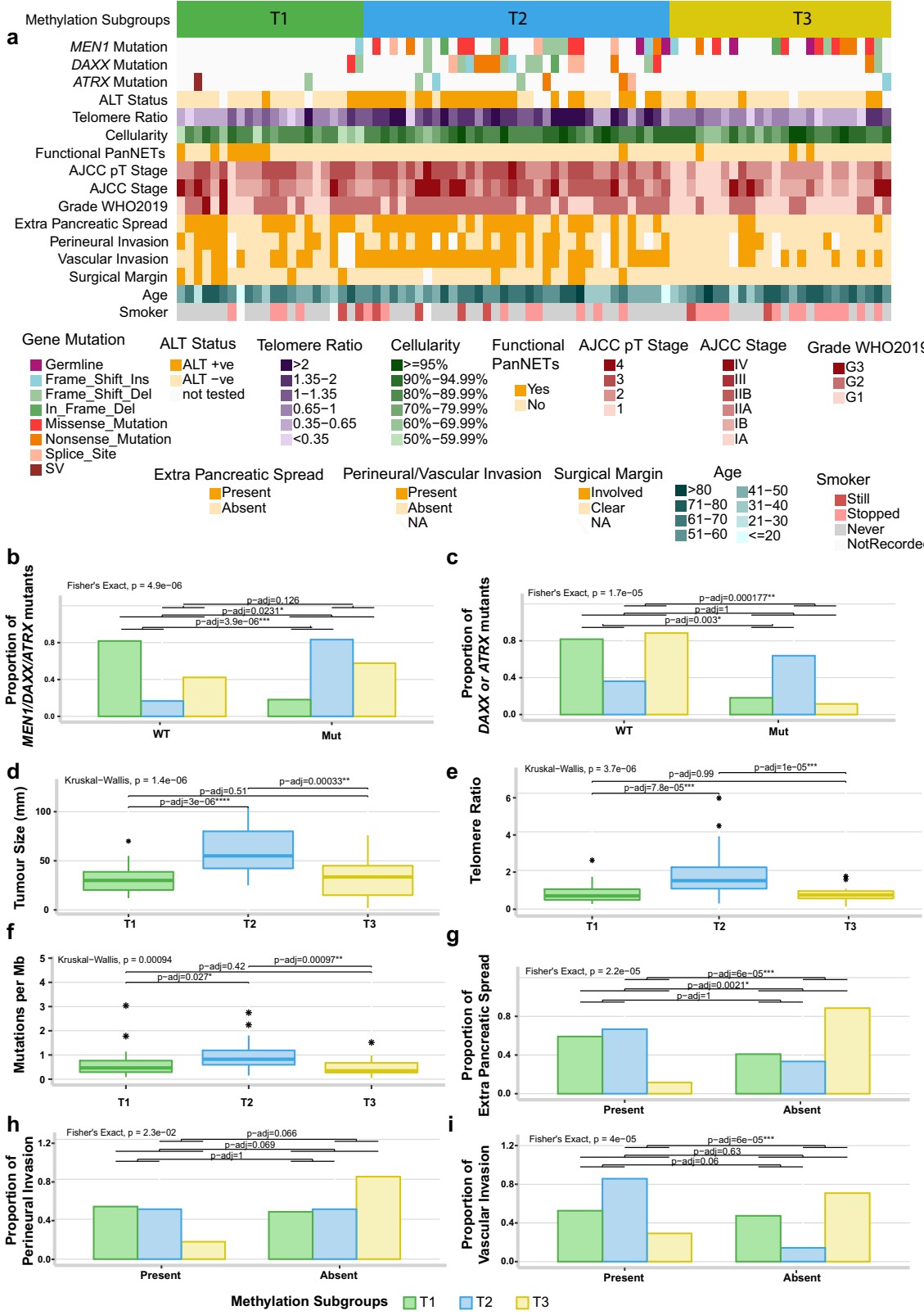

of the 6861 (out of 411,159 CpG sites) most variable CpG sites ($\beta$-value SD $\geq 0.20$ across tumours) located within promoter regions in those chromosomes that presented recurrent LOH in subgroup T2 (Supplementary Fig. 3). We also evaluated the 7316 most variable probes ($\beta$-value SD $\geq 0.2$ across tumours) located in the body of genes (Supplementary Fig. 4). We did not observe lower levels of methylation of CpG sites in tumours in the T2 subgroup,

which harboured most tumours with recurrent loss or LOH across 11 chromosomes, compared to the levels of methylation in the other subgroups (T1 and T3).

**Differentially methylated genes between subgroups and correlation with gene expression.** Next, we compared the 3 subgroups (identified by 3378 most variable probes across tumours not

**Fig. 1 PanNET subgroups identified by methylation profile and unsupervised hierarchical clustering across 84 primary tumours.** Tumours (columns) are presented in the same order as dichotomized clustering presented in Supplementary Fig. 1. **a** Relationship of subgroups with genomic and clinical features. **b** T1 subgroup shows significant enrichment of tumours wild-type for *MEN1/DAXX/ATRX* genes (Fisher's exact test). **c** T2 subgroup tumours are enriched for mutations in *ATRX* and *DAXX* genes (Fisher's exact test). **d** T2 subgroup tumours were larger than those in the other two subgroups (Wilcoxon's rank-sum test). **e** T2 subgroup tumours had significant longer telomeres (Wilcoxon's rank-sum test). **f** T2 subgroup tumours harboured more mutations per megabase (Mb) than tumours in other subgroups (Wilcoxon's rank-sum test). **g** T3 subgroup tumours presented significant less extra-pancreatic spread than tumours in the other subgroups (Fisher's exact test). **h** T3 subgroup tumours had a trend towards less perineural invasion. **i** T3 subgroup tumours had less vascular invasion than the other two subgroups. ALT status: alternative lengthening of telomeres assessed using C-Tailing qPCR[2]; ALT+ve: positive for ALT, ALT−ve: negative for ALT; Telomere Ratio: reads with telomeric repeats were counted in both the tumour and matched normal sample and normalized to the mean genomic coverage of the sample using qMotif[2] for both the tumour and matched normal sample and the ratio gives us an indication of shortening or lengthening in relation to normal sample. Functional PanNETs: tumours that overproduce biologically active hormone. The box within the boxplots represents a range of values from the first to third quantile and the line within represents the median value of the distribution. The whiskers represent the maximum and minimum values of the distribution excluding outliers and an asterisk represents any outlier.

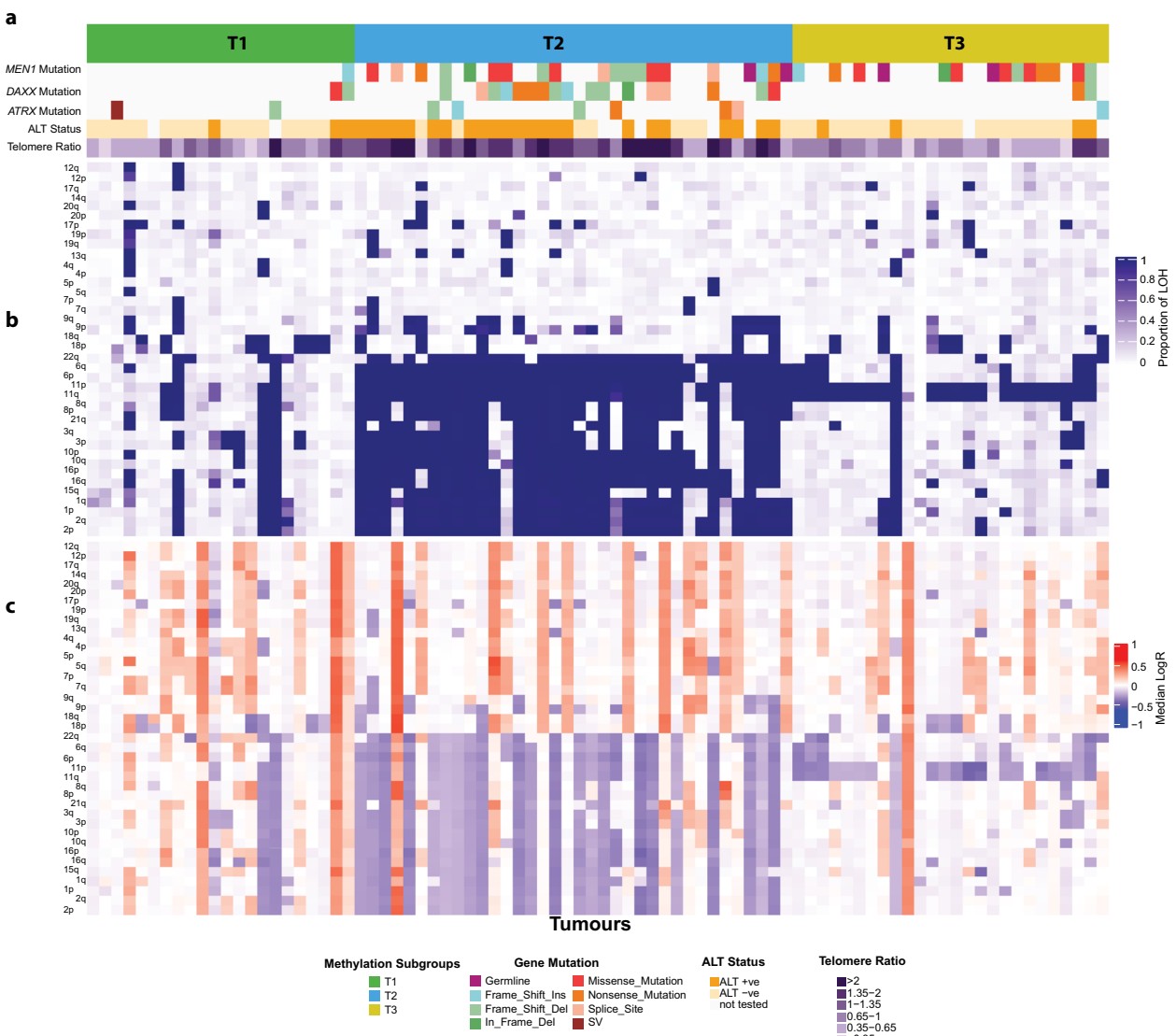

**Fig. 2 Association of methylation subgroups with genomic features.** Tumour (columns) are presented in the same order as dichotomized clustering presented in Supplementary Fig. 1. **a** Mutations in *ATRX, DAXX* and *MEN1* genes, alternative lengthening of telomeres (ALT) and telomere length. **b** Proportion of the chromosomes arms affected by LOH. **c** Median logR ratio across chromosome arms as a reference to copy number changes (by providing an estimate of the allelic ratio). Chromosomal arm copy number and LOH (rows) were clustered using Euclidian distances and Ward's clustering method.

methylated in normal adjacent pancreata) to identify differentially methylated sites across the entire genome and gain some potential insight into the biology that may differentiate the groups. We identified 11,042 GpG sites (out of 411,159) differentially methylated between at least one pair of subgroups. Of these, 3603 mapped to promoter regions of genes and 7439 mapped elsewhere in the genome. A total of 6185 CpG sites were differentially methylated between subgroup T1 and T2 ($q < 0.05$ and a difference

of average $\beta$-value $\geq 0.20$), with 1884 of those CpGs in gene promoter regions (Supplementary Data 2). CpG sites (4044) were differentially methylated between T1 and T3 (with 1563 CpGs in promoter regions, Supplementary Data 3), and 3586 between T2 and T3 (with 1113 CpGs in promoter regions, Supplementary Data 4). When evaluating the number of genes affected by differentially methylated CpG sites, 2630 genes harboured CpG sites differentially methylated between subgroups T1 and T2 with 1401 genes, with promoter regions differentially methylated. When comparing methylation between T1 and T3, 1907 genes harboured differentially methylated CpG sites with 1025 genes, with promoter differentially methylated. Between T2 and T3, 1731 genes harboured sites differentially methylated with 834 of those with probes in the CpG "CpG sites" promoter region (Supplementary Data 5).

We then performed a correlation of methylation levels and gene expression for the 47 cases where RNA sequencing (RNASeq) data were available. CpG sites that mapped to genes and were differentially methylated between at least one pair of subgroups were tested with the aim to gain insights into potential biological mechanisms that could be impacted by aberrant methylation. From the 11,042 CpG sites differentially methylated between at least 1 pair of subgroups, 7303 CpG sites mapped to the footprint of 4317 genes (regions TSS1500, TSS200, 5′-UTR, First Exon, gene body or 3′-UTR). Out of those genes, 987 genes were considered expressed below the detection limits in all 47 cases (876 genes had <3 counts per million (CPM) and 111 had no record in RNASeq data) and were excluded from correlation estimates. A total of 1573 CpG sites showed significant correlation between methylation levels and the gene expression of 910 genes (Supplementary Data 5). A total of 35 genes had 5 or more differentially methylated CpG sites with significant correlation with the expression of the gene, 12 out 35 genes are transcription regulators (Supplementary Data 6). Correlations were positive and negative, depending on the gene and location of differentially methylated probes in the gene. Details of gene expression and methylation in each subgroup are compiled in Supplementary Data 6. Three out of 35 genes that showed significant correlation between the expression and methylation have been previously reported to be aberrantly methylated in subgroups of PanNETs: O6-methylguanine-DNA methyltransferase (MGMT)[21], Pancreatic and Duodenal Homeobox 1 (PDX1)[4] and caspase 8 (CASP8)[12,14].

Gene body methylation of MGMT was significantly lower in tumours in the subgroup T2 compared to tumours in the other groups (T1 and T3). T2 tumours also harboured recurrent loss of chromosome 10q arm including the loci of MGMT (Fig. 3a). In our cohort, 46 CpG sites mapped to the MGMT gene (45 gene body and 1 to promoter region) were differentially methylated between the subgroups and were positively correlated with the gene expression (Supplementary Data 5 and Fig. 3b). Only one CpG site (cg24420981) located in the promoter region of MGMT was differentially methylated between at least one pair of subgroups and unexpectedly also showed a positive correlation with expression. Gene body methylation and expression of MGMT was lower in tumours within subgroup T2 compared to tumours grouped in T1 and T3 as MGMT methylation was not differentially methylated between these subgroups (Fig. 3c, d). Low methylation in the gene body together with LOH could potentially drive low expression of MGMT within the T2 subgroup. We investigated the levels of methylation of the two genes flanking MGMT, Protein Tyrosine Phosphatase Receptor Type E (PTPRE) and EBF Transcription Factor 3 (EBF3), and did not observe the loss of gene body methylation in tumours in the T2 subgroup (Supplementary Data 6), suggesting gene-specific loss of gene body methylation of MGMT.

The PDX1 gene had 19 CpG sites differentially methylated between the 3 subgroups, with 6 of those presenting a significant negative correlation with gene expression (Supplementary Data 5 and Supplementary Fig. 7a). CpG sites with correlation to RNASeq data were located, at TSS1500 ($n = 2$), in the body ($n = 2$) and at 3′-UTR ($n = 2$) of the gene (Supplementary Fig. 7b, c). In our cohort, the PDX1 gene presented hypermethylated CpG sites in the tumours in subgroups T2 and T3 (mutant ATRX/DAXX/MEN1) compared to tumours in the subgroup T1, which are mainly wild-type for those genes and this subgroup was enriched for functional tumours (Supplementary Fig. 7d, e). There was only a trend for higher gene expression of PDX1 in subgroup T1 (Supplementary Fig. 7f). These results need to be treated with caution, as we had gene expression for only 47 cases (8 cases in the T1 subgroup). Chan et al.[4] suggested two genes that differentiate ATRX/DAXX/MEN1 mutants from wild-type PanNETs, Aristaless-related homeobox (ARX) and PDX1. As ARX gene is located in the X chromosome, which is normally removed from methylation analysis, we evaluated ARX methylation patterns separately. We observed no differences in the methylation levels between the groups (Supplementary Fig. 8) even when evaluating males and females separately (Supplementary Data 7). However, we observed a higher gene expression in T2 and T3 compared to T1 (wild-type tumour). This is in agreement with previous studies[4,22], and suggests that the methylation status of ARX may not affect expression of the gene. PDX1 had lower expression and hypermethylation in ATRX/DAXX/MEN1 mutant tumours (T2 and T3) than wild-type tumours (T1). Moreover, most samples with strong hypomethylation of PDX1 in subgroup T1 were functional tumours (Supplementary Fig. 7a). PDX1 expression was previously suggested to be a marker of pancreatic $\beta$-cells[23]. We investigated the gene expression profile of pancreatic $\alpha$- and $\beta$-cell types using the marker genes described by Muraro et al.[23] (Supplementary Fig. 9). Despite the small number of cases, our data suggest that subgroup T1 tumours have higher expression of markers associated with $\beta$-cells supporting a previous report that these tumours might have originated from pancreatic $\beta$-cells.

The identification of the cell of origin was not the primary aim of the present study, but has been integral to recent studies[15–17] that have identified similarities of PanNETs with $\alpha$- or $\beta$-islet cells and pathways of tumorigenesis. With this in mind, we compared five islet cells (two $\alpha$- and three $\beta$-islet cells)[24] with our subgroups. Supplementary Fig. 10 shows that methylation of a set of probes with most variable methylation between $\alpha$- and $\beta$-islet cells, suggests T1 tumours to be $\beta$-like tumours, and T2 and T3 tumours to be $\alpha$-like tumours in agreement with gene expression profiles in the subset of 47 tumours with RNASeq and recent publications.

## Discussion

This study reports the genome-wide patterns of methylation of sporadic primary PanNETs. We identified three PanNETs subgroups, which showed relationships of the methylation patterns with other genomic and clinical features, providing novel insights into mechanisms driving this heterogeneous disease with potential clinical implications.

Previous studies have suggested the potential of DNA methylation to identify PanNETs subgroups; however, they have used a smaller number of DNA methylation sites[3,12,14], smaller cohorts[7] or mixed cohorts of primary and metastatic tumours[4], and they did not explore associations with clinical and/or other genomic features. The subgroups identified in the present study based on DNA methylation profiles are in line with those described by Lawrence et al.[5] based on copy number and somatic mutations, and highlight the association between methylation status, copy

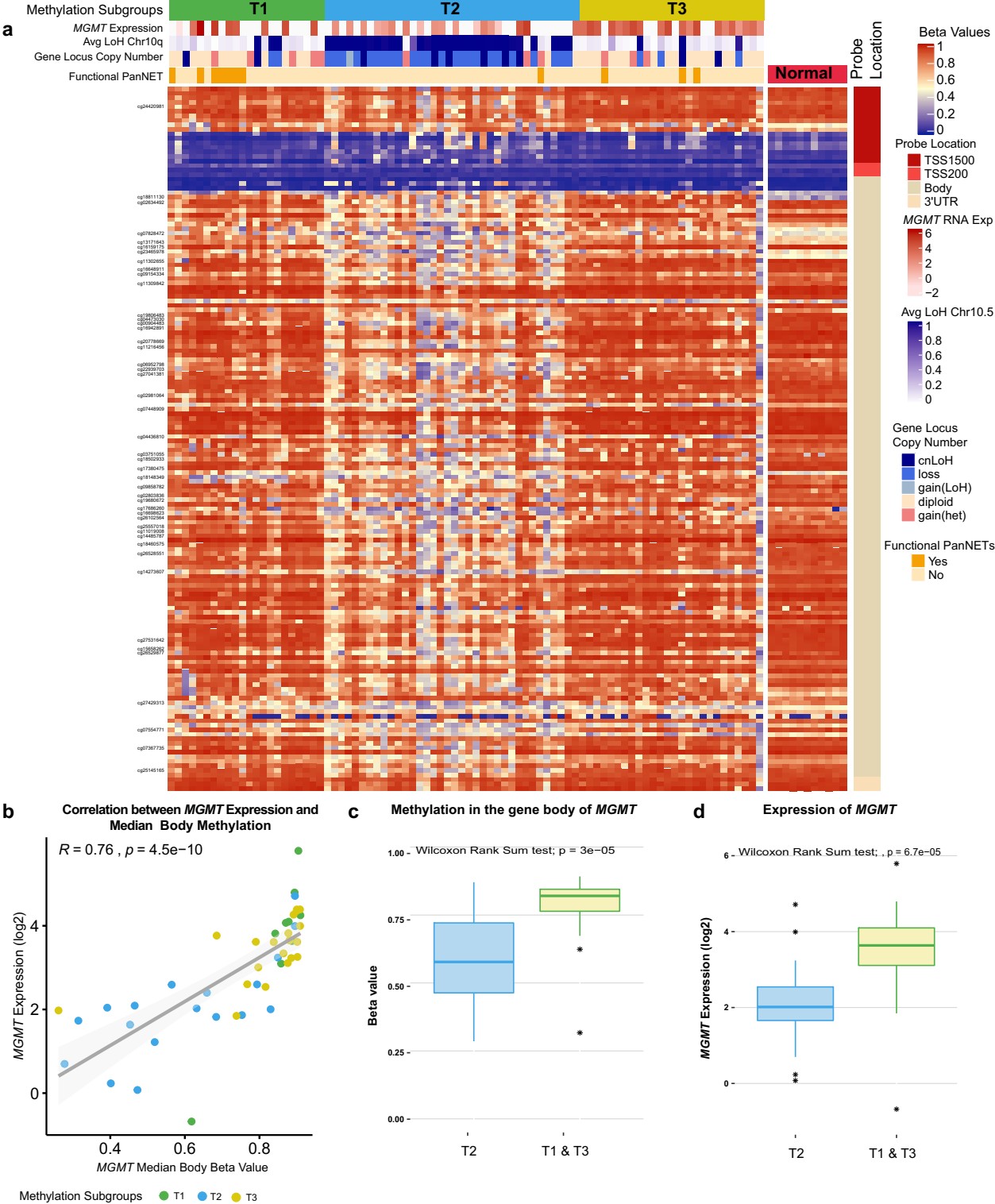

**Fig. 3 Methylation levels of the *MGMT* gene.** Tumour (columns) are presented in the same order as dichotomized clustering presented in Supplementary Fig. 1. **a** Heat map showing methylation levels across 156 CpG sites mapped to the *MGMT* gene, which passed the filters. Probes are plotted by genomic coordinates from 5′ to 3′ direction. Forty-six CpGs sites indicated on the left were differentially methylated between tumour subgroups and correlated with *MGMT* gene expression as assessed in 47 cases with RNASeq data. Levels of methylation in normal adjacent pancreata of CpG sites mapped to the *MGMT* gene is presented on the right. **b** Correlation of methylation median levels of 44 CpG sites located in the *MGMT* gene body and *MGMT* expression in the 47 cases with RNASeq data. **c** Average levels of methylation of *MGMT* gene body CpG sites for tumours in subgroup T2 (*n* = 36) vs. subgroups T1 and T3 (*n* = 48). **d** Gene expression levels of *MGMT* gene across 47 cases with RNASeq data (T2 *n* = 17, T1 and T3 *n* = 30). The box within the boxplots represents a range of values from the first to third quantile and the line within represents the median value of the distribution. The whiskers represent the maximum and minimum values of the distribution excluding outliers and an asterisk represents any outlier.

number alterations and somatic mutations, adding another important element to our knowledge of the genetic landscape of these neoplasms.

The subgroup T1 included tumours that were mainly wild-type for the commonly mutated genes *ATRX*, *DAXX* and *MEN1*. This subgroup presented tumours with heterogeneous patterns of copy number alteration, agreeing with Chan et al.[4] who reported gene expression and methylation profiles to be more homogenous in mutant *ATRX/DAXX/MEN1* tumours than in wild-type tumours. Subgroup T1 was also enriched for functional tumours (7 of 11 represented in the cohort).

Tumours in subgroup T2 harboured more mutations in *ATRX*, *DAXX* and *MEN1* with recurrent loss/LOH across 11 chromosomes. This is the first study that highlights a correlation of DNA methylation profiles with mutations affecting *DAXX/ATRX* genes and recurrent chromosomal losses, which have been associated with reduced survival in PanNETs[2,3,5,25]. We investigated the levels of methylation at CpG sites associated with genes mapping in regions of recurrent loss and LOH and we did not observe lower levels of methylation in those regions in tumours harbouring recurrent loss compared to other subgroups, suggesting that together LOH and methylation could potentially affect tumorigenesis in PanNETs. Tumours in this subgroup were larger and harboured more somatic point mutations than the other two subgroups. Lawrence et al.[5] have suggested that haploinsufficiency by somatic LOH could be a plausible mechanism contributing to PanNETs development by affecting gene expression of a range of tumour suppressors. Our data suggest an extra layer of complexity, by an orchestrated alteration of patterns of methylation that could potentially explain the similarity of the subgroups observed in the present study and that of Lawrence et al.[5], supporting the hypothesis of a joint mechanism (copy number/LOH plus methylation) driving tumorigenesis in this cancer.

T3 tumours harboured mutations in *MEN1* and had no major recurrent copy number changes in their genomes except loss or LOH of the chromosome 11. T3 tumours had a lower incidence of perineural, vascular invasion and extra-pancreatic spread, and a higher proportion of grade G1 tumours compared to the other groups (T1 and T2), suggesting that these tumours have less aggressive behaviour[10]. *MEN1* mutations also play a role in inherited PanNETs. A previous study[7] evaluating methylation of nine sporadic and ten inherited *MEN1*-related PanNETs suggested that *MEN1* mutated tumours in both settings are more similar than *VHL* inherited tumours ($n = 10$). However, sporadic and inherited MEN1-related PanNETs have distinct patterns of methylation. It would be interesting in the near future to evaluate how *MEN1*-related inherited PanNETs compare to the two subgroups, harbouring *MEN1* mutations identified here, which have distinct histological parameters indicative of prognosis.

Overall, the results presented here advance the comprehension of the genetic and epigenetic landscape of PanNETs, indicating also that patterns of methylation have the potential to stratify PanNETs prognosis. Our findings are the initial report of the potential for DNA methylation as a biomarker in PanNETs, which needs to be validated in other cohorts. If validated, the next step towards the use in clinical practice is the identification of a smaller number of CpG sites with PanNET-specific methylation compared to other tissues that can specifically distinguish the subtypes and could in the future influence how the patients are managed in the clinic for treatment and surveillance.

*PDX1* presented hypermethylation and lower expression in tumours with mutations in the *ATRX/DAXX/MEN1* genes (T2 and T3) than in wild-type tumours (T1). This is in agreement with findings from Chan et al.[4], supporting the trend in gene expression observed in our study. *PDX1* is a transcriptional factor for several genes, including insulin, somatostatin, glucokinase, islet amyloid polypeptide, and glucose transporter type 2. *PDX1* plays a major role in glucose-dependent regulation of insulin gene expression. Interestingly subgroup T1 was enriched for the majority of functional tumours in the cohort including five of six insulinomas. Boons et al.[15] have used methylation probes in the *PDX1* loci to identify two PanNETs subtypes (A and B), which they reported similar to α- and β-islet cells, respectively. Subtype A had a significant worse prognosis compared to subtype B, with most insulinomas in the latter group. Subtype B was mostly wild-type for *ATRX/DAXX/MEN1* genes, which is in agreement with our T1 subgroup. In our study, we further identified two groups with different clinical and genomic features. Gene expression and methylation patterns of these groups (T2 and T3) suggests potential similarity with α-islet cells. In Boons et al.[15], subtype A seems to have a further branching in their cluster analysis that suggests a potential difference in metastatic disease, but the authors did not further discuss that. Here we speculate that further subgrouping in their subtype A could be related to the two groups (T2 and T3) seen in our study. T2 and T3 mostly are non-functional PanNETs but showed significant differences in histological parameters indicative of prognosis, with T2 associated with markers of worse prognosis.

Tumours in the T2 subgroup had lower methylation in CpGs located in the body of *MGMT* gene, which showed positive correlation with gene expression. These data are in agreement with previous studies, suggesting that *MGMT* gene body methylation has a role in *MGMT* expression[26,27], and places this phenomenon in a specific subgroup of PanNETs. Lower expression of *MGMT* has been suggested to increase tumour sensitivity to Temozolomide, which is frequently used for treatment of advanced PanNETs. However, studies have presented contradictory results between expression of *MGMT* and response to Temozolomide in PanNETs[21,28,29]. A clinical trial for well-differentiated advanced duodeno-pancreatic, lung, or unknown primitive NETs is currently evaluating if *MGMT* promoter methylation could predict response to alkylating agents (NCT03217097)[30]. In our study, only one CpG site located in the promoter region of *MGMT* was differentially methylated between the subgroups, and unexpectedly showed a positive correlation with expression. The role of methylation in the body of genes and its relationship with gene expression is not fully understood. Studies have suggested potential mechanisms such as regulation of alternative promoters, regulation of retrotransposon elements influencing alternative transcription and regulation of other functional elements that maintain efficiency of transcription[31–33]. *MGMT* gene body methylation would require further investigation due to its potential clinical impact as a predictive marker for treatment of PanNETs.

In summary, this study reports a genome-wide scan of DNA methylation in PanNETs, providing evidence that aberrant DNA methylation plays an important role in their tumorigenesis. This is the largest cohort with integration of methylation with genomic and clinical information suggesting that an orchestrated epigenetic deregulation plays a role together with somatic LOH/copy number changes in this disease. DNA methylation might contribute to the heterogeneity in clinical presentation and behaviour of these tumours. Our findings may also have future clinical implications for stratifying the prognosis and assisting therapeutic choices for PanNETs patients.

## Methods

**Cohort and samples**. A total of 84 clinically sporadic primary PanNETs and 11 normal pancreata samples from cancer patients were evaluated for their whole-genome DNA methylation profile using the Illumina 450 K arrays. Tissue processing and DNA extraction were as previously described[34]. Methylation data were generated using the same DNA extraction that was used for the whole-genome

landscape of these tumours that has previously been characterized by Scarpa et al.[2]. Their study included a description of the tumour content, copy number profiles, somatic and germline mutations, together with clinical parameters. Here we used this publicly available information to complement analysis of methylation patterns across tumours.

**Ethics**. Written informed consent was obtained from all patients enroled in this study, with approval from:

*ARC-Net, University of Verona*. Approval number 1885 from the Integrated University Hospital Trust (AOUI) Ethics Committee (Comitato Etico Azienda Ospedaliera Universitaria Integrata) approved in their meeting of 17 November 2010, and documented by the ethics committee 52070/CE on 22 November 2010 and formalized by the Health Director of the AOUI on the order of the General Manager with protocol 52438 on 23 November 2010. The specific study was approved with approval (number 2173) by the Ethics Committee with protocol 25979 dated 29/95/2012 and ratified by the Health Director of the AOUI with protocol 26775 dated 01/06/2012.

*Australian Pancreatic Genome Initiative. Central approval*. Sydney Local Health District Human Research Ethics Committee (X16-0293); Royal Adelaide Hospital Human Research Ethics Committee (091107a); Metro South Human Research Ethics Committee (09/QPAH/220); South Metropolitan Area Health Service Human Research Ethics Committee (09/324); Southern Adelaide Health Service/ Flinders University Human Research Ethics Committee (167/10); The University of Queensland Medical Research Ethics Committee (2009000745); Greenslopes Private Hospital Ethics Committee (09/34); North Shore Private Hospital Ethics Committee (2016/016).

*Baylor College of Medicine*. Institutional Review Board protocol numbers H-29198 (Baylor College of Medicine tissue resource), H-21332 (Genomes and Genetics at the BCM-HGSC) and H-32711 (Cancer Specimen Biobanking and Genomics).

*QIMR Berghofer Medical Research Institute*. P3462 (QIMR Berghofer Human Research Ethics Committee).

**Bisulfite conversion, 450 K methylation arrays**. Genomic DNA (500 ng) was bisulfite converted using EZ DNA methylation Kits (Zymo Research) according to the manufacturer's protocol for Illumina Methylation arrays. Bisulfite converted DNA was whole-genome amplified and hybridized to Infinium Human Methylation 450 K BeadChips (Illumina) according to the manufacturer's protocol. Arrays were scanned using an iScan (Illumina).

**DNA methylation analysis**. Raw IDAT files were imported[35], filtered and normalized using the ChAMP[36] package implemented in R. Probes were primarily filtered out if detection $p$-value > 0.01 or there were fewer than three beads in at least 5% of samples. A $\beta$-mixture quantile normalization (BMIQ)[37] was performed to account for probe type1 and type 2 biases, followed by a quantile normalisation (QN). Further filtering was performed to remove probes in non-CpG sites, X or Y chromosome, single-nucleotide polymorphism-related polymorphisms as per Zhou et al.[38] and probes that map to multiple locations as per Nordlund et al.[39] (Supplementary Table 1). After the filtering process, 411,159 probes were used for further analysis. A single value decomposition[40] analysis determined no significant batch effects (Supplementary Fig. 11). Tumour methylation arrays were provided by the International Cancer Genome Consortium (ICGC, https://icgc.org/) under projects PAEN-AU and PAEN-IT. DNA methylation array data generated for normal adjacent samples was deposited in Gene Expression Omnibus (GSE149395).

**Statistics and reproducibility**. To identify CpG sites (CG positions in the genome assayed by arrays) with potential cancer-specific DNA methylation, we selected probes in promoter regions (indicated in the vendor annotation file as TSS1500, TSS200, 5′-UTR or 1stExon). Next, we identified CpG sites located in promoter regions with low levels of methylation ($\beta$-value < 0.3) in all normal adjacent pancreata and from those selected the most variable probes with a SD > 0.20 of DNA methylation levels across all tumours. The CpG sites with the most variable methylation levels across tumours were then dichotomized (representing a presence $\beta$-value $\geq$ 0.3 or absence $\beta$-value < 0.3 of methylation) and clustered to obtain potential subgroups of tumour samples. The unsupervised clustering used a binary distance measure and Ward's clustering method. The approaches described above were taken to minimize potential confounding signal from non-tumour cells in tumour samples. The differential methylation analyses between subgroups identified by clustering were performed using $t$-tests for 411,159 probes with adjustment for multiple testing using Benjamini and Hochberg method using the R function p. adjust. Significance was defined if the adjusted $p$-value < 0.05 and the difference in the average $\beta$-values between subgroups > 0.2. Differential methylation analysis between identified subgroups aimed to gain biological insights about the differences in those groups. An investigation to identify potential associations was performed

on the subgroups identified from methylation data with clinical and genomic features. For continuous variables, Kruskal–Wallis tests with post hoc analyses were performed using a Wilcoxon's rank-sum test and $p$-values were adjusted for multiple testing using Benjamini and Hochberg method with the R function p. adjust. For the categorical variables, Fisher's exact tests were performed to compare the relative proportions of the variables between the subgroups. R (version 3.5.1) was used for all analyses and visualizations. Differential methylation between five islet cell[24] and each subgroup was performed, and probes with an average difference > 0.3 between $\alpha$- and $\beta$-islet cells were compared in a heat map to give potential insight about cell of origin of the subgroups identified.

*RNA sequencing*. RNASeq data were available for a subset of 47 cases in the present cohort. RNASeq data of 27 cases were previously published by Scarpa et al.[2] and a new extra set of 20 cases formed the set of 47 cases with gene expression data under https://www.ebi.ac.uk/ega/search/site/EGAD00001006063. Sequence reads were trimmed for adapter sequences using Cutadapt (version 1.9)[41] and aligned using STAR (version 2.5.2a)[42] to the GRCh37 human reference genome assembly using the gene, transcript and exon features model of Ensembl (release 70). Quality-control metrics were computed using RNA-SeQC (version 1.1.8)[43] and transcript abundances were quantified using RSEM (version 1.2.30)[44]. Further analyses of the RNASeq data were carried out in R (version 3.5.1). Genes were considered expressed if they had 3 CPM in at least 5% of cases ($n = 2$). Trimmed mean of $M$-values normalization was performed using the edgeR package[45]. The batch effect between the two RNASeq data sets (27 published and 20 new cases) was corrected using ComBat from the "sva" R package (version 3.30.1)[46] (Supplementary Fig. 12). Correlations between methylation and gene expression were calculated using a Pearson's correlation and $p$-values were adjusted for multiple testing using Benjamini–Hochberg method using the R function p.adjust.

**Reporting summary**. Further information on research design is available in the Nature Research Reporting Summary linked to this article.

## Data availability
Methylation data used in this study has been deposited in the NCBI Gene Expression Omnibus (GEO) under the accession number GSE149395 and ICGC portal (International Cancer Genome Consortium (ICGC) https://icgc.org/ under projects PAEN-AU and PAEN-IT). RNASeq data are available on EGA (https://www.ebi.ac.uk/ega/search/site/EGAD00001006063). Source data for figures presented in the manuscript are available as Supplementary Data 8.

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

## Acknowledgements

We thank all patients and funding below that made this study possible. Associazione Italiana Ricerca Cancro (AIRC 5×1000 n. 12182 and Start up n. 18718); Fondazione Italiana Malattie Pancreas–Ministero Salute [FIMPCUP_J38D19000690001]; Fondazione Cariverona: Oncology Biobank Project "Antonio Schiavi" (prot. 203885/2017); European Community ERANET PMTR-pNET, cod. D18TR5, B46C17000260001; National Health and Medical Research Council of Australia (NHMRC; 631701, 535903, CDF 1112113, PRF 1025427, SRF 455857, 535903); The Queensland State Government Smart State National and International Research Alliances Program (NIRAP); Institute for Molecular Bioscience/University of Queensland; Australian Government: Department of Innovation, Industry, Science and Research (DIISR); Australian Cancer Research Foundation (ACRF); Cancer Council NSW (SRP06-01, SRP11-01. ICGC); Cancer Institute NSW (10/ECF/2-26; 06/ECF/1-24; 09/CDF/2-40; 07/CDF/1-03; 10/CRF/1-01, 08/RSA/1-15, 07/CDF/1-28, 10/CDF/2-26,10/FRL/2-03, 06/RSA/1-05, 09/RIG/1-02, 10/TPG/1-04, 11/REG/1-10, 11/CDF/3-26); Garvan Institute of Medical Research; Avner Nahmani Pancreatic Cancer Research Foundation; R.T. Hall Trust; Petre Foundation; Philip Hemstritch Foundation; Gastro-enterological Society of Australia (GESA Senior Research Fellowship); Royal Australasian College of Surgeons (RACS); Royal Australasian College of Physicians (RACP); Royal College of Pathologists of Australasia (RCPA); QIMR Berghofer Medical Research; The Keith Boden Fellowship supporting KN; NHGRI U54 HG003273; CPRIT grant RP101353-P7; Wellcome Trust Senior Investigator Award (103721/Z/14/Z); CRUK Programme (C29717/A17263 and C29717/A18484); CRUK Glasgow Centre (C596/A18076); CRUK Clinical Training Award (C596/A20921); Pancreatic Cancer UK Future Research Leaders Fund; The Howat Foundation; University of Glasgow. N.W. was supported by National Health and Medical Research Council of Australia (NHMRC; SRF 1139071).

## Author contributions

V.L., K.N., A.S., N.W., A.V.B., and S.M.G.: concept and design. K.N., V.L., A.S., N.W., R.T.L., C. Luchini, and V.C.: writing team. V.L., K.N., F.N., and A.M.P.: methylation analysis. V.L., L.T.K., R.L.J., R.T.L., A.M.P., A.M., A. Sadanandam, D.C., and P.B.: RNASeq data and analysis. V.L., K.N., APGI, ARC-Net, B.R., C. Leonard, S.W., K.N., A.M.P., R.T.L., A.M., M. Milella, F.D., R.S., S.C., L.L., P.P., N.B.J., M.C.G., D.M.M., D.A.W., R.A.G., J.S.S., A.J.G., A.L.J., J.V.P., A.V.B., S.M.G.: sample collection, processing, quality control and clinical annotation, and pathology assessment. A.S., M. Milione, A.V.B., and S.M.G.: attracted funding. All authors have read and approved the final manuscript.

## Competing interests

N.W. and J.V.P are co-founders and Board of genomiQa pty ltd. The other authors declare no competing interests.

## Additional information

[1]QIMR Berghofer Medical Research Institute, 300 Herston Road, Brisbane, Queensland 4006, Australia. [2]ARC-Net Centre for Applied Research on Cancer, University and Hospital Trust of Verona, Verona 37134, Italy. [3]Division of Molecular Pathology, The Institute of Cancer Research, London, UK. [4]The Royal Marsden Hospital, London, UK. [5]Section of Pathology, Department of Diagnostics and Public Health, University of Verona, Verona, Italy. [6]ENETS Center of Excellence, University and Hospital Trust of Verona, Verona, Italy. [7]Section of Oncology, Department of Medicine, University and Hospital Trust of Verona, Verona, Italy. [8]The Pancreas Institute, University and Hospital Trust of Verona, Verona, Italy. [9]Wolfson Wohl Cancer Research Centre, Institute of Cancer Sciences, University of Glasgow, Garscube Estate, Switchback Road, Bearsden, Glasgow G61 1BD, UK. [10]Academic Unit of Surgery, School of Medicine, College of Medical, Veterinary and Life Sciences, University of Glasgow, Glasgow Royal Infirmary, Glasgow G4 0SF, UK. [11]West of Scotland Pancreatic Unit, Glasgow Royal Infirmary, Glasgow G31 2ER, UK. [12]Department of General Surgery, University of Heidelberg, Im Neuenheimer Feld 110, 69120 Heidelberg, Baden-Württemberg, Germany. [13]Department of Pathology, Queen Elizabeth University Hospital, Greater Glasgow & Clyde NHS, Glasgow G51 4TF, UK. [14]Department of Molecular and Human Genetics, Human Genome Sequencing Center, Baylor College of Medicine, One Baylor Plaza, MS226, Houston, Texas 77030-3411, USA. [15]Michael E. DeBakey Department of Surgery and The Elkins Pancreas Center, Baylor College of Medicine, One Baylor Plaza, Houston, Texas 77030-3411, USA. [16]Human Genome Sequencing Center, Baylor College of Medicine, Houston, Texas 77030, USA. [17]Department of Pathology, Fondazione IRCCS Istituto Nazionale dei Tumori di Milano, Milan, Italy. [18]Ospedale Pederzoli, Peschiera del Garda, Verona, Italy. [19]University of Sydney, Sydney, New South Wales 2006, Australia. [20]The Kinghorn Cancer Centre, Garvan Institute of Medical Research, 370 Victoria Street, Darlinghurst, Sydney, New South Wales 2010, Australia. [21]University of Melbourne Centre for Cancer Research, Victorian Comprehensive Cancer Centre, 305 Grattan Street, Melbourne, Victoria 3000, Australia. [22]Institute for Molecular Bioscience, The University of Queensland, St Lucia, Brisbane, Queensland 4072, Australia. [23]Royal North Shore Hospital, Westbourne Street, St Leonards, New South Wales 2065, Australia. [24]Bankstown Hospital, Eldridge Road, Bankstown, New South Wales 2200, Australia. [25]Liverpool Hospital, Elizabeth Street, Liverpool, New South Wales 2170, Australia. [26]St Vincent's Hospital, 390 Victoria Street, Darlinghurst, New South Wales 2010, Australia. [27]Westmead Hospital, Hawkesbury and Darcy Roads, Westmead, New South Wales 2145, Australia. [28]Royal Prince Alfred Hospital, Missenden Road, Camperdown, New South Wales 2050, Australia. [29]Prince of Wales Hospital, Barker Street, Randwick, New South Wales 2031, Australia. [30]Fremantle Hospital, Alma Street, Fremantle, Western Australia 6959, Australia. [31]Epworth HealthCare, 89 Bridge Road, Richmond, Victoria 3121, Australia. [32]Royal Adelaide Hospital, North Terrace, Adelaide, South Australia 5000, Australia. [33]Flinders Medical Centre, Flinders Drive, Bedford Park, South Australia 5042, Australia. [34]Envoi Pathology, 1/49 Butterfield Street, Herston, Queensland 4006, Australia. [35]Princess Alexandra Hospital, Cornwall Street & Ipswich Road, Woolloongabba, Queensland 4102, Australia. [36]Austin Hospital, 145 Studley Road, Heidelberg, Victoria 3084, Australia. [37]Johns Hopkins Medical Institute, 600 North Wolfe Street, Baltimore, Maryland 21287, USA. [38]These authors jointly supervised this work: Katia Nones, Aldo Scarpa. *Lists of authors and their affiliations appear at the end of the paper. ✉email: Katia.Nones@qimrberghofer.edu.au; aldo.scarpa@univr.it

## APGI

**Garvan Institute of Medical Research** Lorraine A. Chantrill[9,18], Paul Timpson[20], Angela Chou[20,21], Marina Pajic[20], Angela Murphy[20], Tanya Dwarte[20], David Hermann[20], Claire Vennin[20], Thomas R. Cox[20], Brooke Pereira[20], Shona Ritchie[20], Daniel A. Reed[20], Cecilia R. Chambers[20], Xanthe Metcalf[20] & Max Nobis[20]

**QIMR Berghofer Medical Research Institute** Pamela Mukhopadhyay[1], Venkateswar Addala[1], Stephen Kazakoff[1], Oliver Holmes[1], Qinying Xu[1] & Scott Wood[1]

**University of Melbourne: Centre for Cancer Research** Oliver Hofmann[21]

**Royal North Shore Hospital** Jaswinder S. Samra[23], Nick Pavlakis[23], Jennifer Arena[23] & Hilda A. High[23]

**Bankstown Hospital** Ray Asghari[24], Neil D. Merrett[24], Darren Pavey[24] & Amitabha Das[24]

**Liverpool Hospital** Peter H. Cosman[25], Kasim Ismail[25] & Chelsie O'Connnor[25]

**St Vincent's Hospital** Alina Stoita[26], David Williams[26] & Allan Spigellman[26]

**Westmead Hospital** Vincent W. Lam[27], Duncan McLeod[27] & Judy Kirk[27]

**Royal Prince Alfred Hospital Chris O'Brien Lifehouse** James G. Kench[28], Peter Grimison[28], Charbel Sandroussi[28] & Annabel Goodwin[25,28]

**Prince of Wales Hospital** R. Scott Mead[9], Katherine Tucker[29] & Lesley Andrews[29]

**Fiona Stanley Hospital** Michael Texler[30], Cindy Forest[30], Mo Ballal[30] & David R. Fletcher[30]

**Epworth Health Care** Nikolajs Zeps[31]

**Royal Adelaide Hospital** Nan Q. Nguyen[32], Andrew R. Ruszkiewicz[32] & Chris Worthley[32]

**Flinders Medical Centre** John Chen[33], Mark E. Brooke-Smith[33] & Virginia Papangelis[33]

**Envoi Pathology** Andrew D. Clouston[34]

**Princess Alexandra Hospital** Andrew P. Barbour[35], Thomas J. O'Rourke[35], Jonathan W. Fawcett[35], Kellee Slater[35], Michael Hatzifotis[35] & Peter Hodgkinson[35]

**Austin Hospital** Mehrdad Nikfarjam[36]

**Johns Hopkins Medical Institutes** James R. Eshleman[37], Ralph H. Hruban[37] & Christopher L. Wolfgang[37]

**University of Glasgow** Judith Dixon[9]

**ARC-Net**
Maria Scardoni[2,5], Claudio Bassi[2,8], Sonia Grimaldi[2], Cinzia Cantù[2,5], Giada Bonizzato[2], Samantha Bersani[5], Davide Antonello[8], Liliana Piredda[2], Nicola Sperandio[2], Stefano Barbi[5] & Paola Merlini[2]

