## [Peer Review File · Communications Biology]

Reviewers' Comments:

Reviewer #1:

Remarks to the Author:

The manuscript by Lakis et al. describes DNA methylation profiling of 84 PanNETs using the Illumina 450K platform. The authors use a specific subset of the data (i.e. using CpG loci that showed so-called "tumor-specific" methylation and mapped to promoter regions of known genes), to cluster the 84 tumors into 3 subtypes. They then went on to look for associations between specific subtypes and (1) the presence of known recurring mutations in PanNETs (e.g. MEN1, ATXR, DAXX), telomere characteristics, copy number variation and LOH, and several histopathological characteristics. They then go on to assess the impact of DNA methylation on gene expression through correlation analysis.

Overall, this study provides some novel information as to the genetic and epigenetic landscape of PanNETs, however two main issues need to be addressed before this manuscript is suitable for publication.

1. The choice of probes (loci) used for the initial clustering of the 3 subtypes is of concern and should be justified by the authors.

a. Probes included in the analysis were only those that did not show significant methylation signal (beta value <0.3) in normal adjacent pancreata samples. This selection criterion has 3 issues: (1) due to field effect, "normal adjacent" pancreatic tissue may or may not be normal; (2) loci that are methylated in normal pancreas but become hypomethylated in tumors would be excluded from the analysis; (3) PanNETs presumably arise from the islets which comprise only 1-2% of pancreatic tissue. Therefore, the vast majority of the methylation signal detected from normal pancreata is from the acinar portion of the tissues as well as vascular tissue, stroma, and immune cells. Thus, the methylation status at a particular CpG locus could be very different in islet cells compared to the other cell types in the normal samples. For example, an unmethylated locus in islet cells would be excluded if the same locus was highly methylated in the exocrine portion of the pancreas.

b. Probes were selected for analysis only if they mapped to the promoter region. Methylation may play a role in inactivation of distal enhancers or other regulatory elements (e.g. insulators) and should not be excluded from analysis for both subtype identification and correlation between methylation and gene expression.

c. It would be worthwhile to rerun the analyses using a more comprehensive probe set.

2. It appears that a significantly larger set of probes was used for the identification of differentially methylated loci (DMLs) between subtypes. DMLs were then discussed in relation to the 3 subtypes identified by a smaller set of probes. This seems odd to this reviewer and the rationale for doing this should be made clear in the Results section.

Minor points:

1. Several terms in the text and/or in the Figure legends should be clarified. For example, the less informed reader shouldn't have to look up the meaning and/or significance of "functional" tumors, ALT, telomere ratio ..

2. In line 184, clarify the meaning of "if the observed methylation profile is driven by copy number," . And in line 189, what does "hypo-methylation" refer to. Hypo-methylation is a relative term. Do the authors mean to say that apparent hypo-methylation may results from copy number loss?

3. Line 211, what is meant by "the footprint of " . Do you mean, the gene body alone, or does it include the promoter regions?

4. In Fig. 4B, the methylation level is more appropriate on the x-axis; same for Suppl. Fig. 6B&C

5. Fig. 4A may be more suitable as a supplemental fig.

6. Line 229-230, Please clarify what is meant by "Low methylation in the gene body together with LOH could potentially drive low expression of MGMT within the T2 subgroup". It's not clear if low

methylation and LOH occur at the same time in a tumor, or independently.

Reviewer #2:

Remarks to the Author:

Lakis et al. report DNA methylation analysis of a large cohort of sporadic PanNETs. The cohort is impressive, the analysis is thorough and very informative and the results are important. The authors are well known for their expertise in PanNET genetics. However, our ability to make robust conclusions from this analysis are limited, mainly due to the first comment, but also in the absence of proper referencing for other recent publications in the field. Here are few points to be addressed in the view of the reviewer:

Major

Adjacent pancreata cannot be used as control for DNA methylation analysis of islet cell tumors, as the former include mainly exocrine pancreas tissue. This selection is very surprising and should be substituted by normal islet cells samples (derived from cadavers, which are also not the perfect choice but much better than practically a different organ). DNA methylation is tissue (or even cell-) specific. Since the entire analysis is based on these tissue samples, the results might be entirely flawed and the analysis should be probably re-done from the square 0.

How much of the data presented and analyzed derived from the data reported in Scarpa et al. Nature 2017?

It is important to assess/validate/reject the data recently reported by Boons, Vandamme et al regarding Pdx1 and Arx promoter methylation and link with tumor aggressivity/prognosis (Cancers, 2020).

Also, please refer to the recent data uploaded to BioRxiv by Di Domenico and co on PanNET methylation analysis.

lines 217-220 – what does DNAm_{et} and gene exp correlation (positive or negative?) reflect? This is a known mechanism of transcription regulation. Is an association with clinical parameters.

Line 221 – lower compared to which group(s)?

Line 224-225 – In line 224 the authors wrote “positive” and probably meant “negative? how do the authors explain positive correlation between promoter methylation and gene expression?

Lines 246-247 – Removing the x chromosome is a reversible action. I strongly suggest to include this analysis.

Minor

60% survival in 5 years: Maybe only for advanced disease, or old data when the PanNET incidentalomas were less frequent. Please add data to support other than ref 1.

Lines 290-291 higher proportion than what?

The figures are of low quality. There are no dendrograms in the heat-maps, was this unsupervised clustering?

Please re-consider whether all figures are contributory and informative.

Reviewer #3:

Remarks to the Author:

Lakis et al report in an international collaboration the correlation between methylation profiles and clinical / genetic data of a large number of PanNETS. They were able to identify three subgroups with distinct patterns of methylation.

They extensively investigated the relation of methylation with commonly mutated genes in these tumors, somatic copy numbers changes and gene expression

In this, to date largest cohort, they were able to show new insights in both genetic and epigenetic landscape of panNETS.

I have a few questions/ suggestions:

- 1) with this strong distinction between the three groups, the authors might give a stronger recommendation how to treat these patients in clinical practice
- 2) the cohort only consist of sporadic tumors. Do the authors feel that their data can be extrapolated to familial PanNETs, e.g. MEN 1 related tumors? Please discuss.
- 3) in the introduction section, in both line 114 and 119, they use the word conversely. They might consider to change one of them.
- 4) discussion section line 297: typo ATRX instead of ATX.

Please find below our responses (blue font) to the reviewer's comments (black font) about the manuscript COMMSBIO-20-1136-T "DNA methylation patterns identify subgroups of pancreatic neuroendocrine tumors with clinical association". The reviewers raised some useful and valid points, and we provide a detailed response and extra analyses to address their comments. We thank the reviewers for their suggestions, as they make our manuscript stronger and clear to the readers.

Reviewers' comments:

Reviewer #1 (Remarks to the Author):

The manuscript by Lakis et al. describes DNA methylation profiling of 84 PanNETs using the Illumina 450K platform. The authors use a specific subset of the data (i.e. using CpG loci that showed so called "tumor-specific" methylation and mapped to promoter regions of known genes), to cluster the 84 tumors into 3 subtypes. They then went on to look for associations between specific subtypes and the presence of known recurring mutations in PanNETs (e.g. MEN1, ATXR, DAXX), telomere characteristics, copy number variation and LOH, and several histopathological characteristics. They then go on to assess the impact of DNA methylation on gene expression through correlation analysis.

Overall, this study provides some novel information as to the genetic and epigenetic landscape of PanNETs, however two main issues need to be addressed before this manuscript is suitable for publication.

1. The choice of probes (loci) used for the initial clustering of the 3 subtypes is of concern and should be justified by the authors.
 - a. Probes included in the analysis were only those that did not show significant methylation signal (beta value <0.3) in normal adjacent pancreata samples.

Response:

Our aim was to identify CpG sites that could stratify tumors with little or no contribution of signal from non-tumor cells. By selecting probes not methylated across all pancreata samples we would expect those CpG sites to have little or no contribution to the methylation signal in the tumor samples. Our thought was that the signal in those sites should be tumor specific, which could then be used to distinguish tumor sub-groups independent of the content of non-tumor cells in the tumor samples. These sites will have little or no confounding signals from other contaminating pancreatic cells (acinar), vascular or immune cells normally present in a clinical sample. The presence of non-tumor cells in tumor samples is not particular to our study, this is a limitation of all cancer methylation studies.

This approach to probe selection, together with the dichotomization of the beta-values for clustering to ameliorate the effect of tumor purity, is not novel, it has been used previously by us and others (see examples below):

- 1) Cancer Genome Atlas Research Network. Comprehensive molecular characterization of gastric adenocarcinoma. *Nature*. 2014; 513:202-209;
- 2) Krause L, et al. (Identification of the CIMP-like subtype and aberrant methylation of members of the chromosomal segregation and spindle assembly pathways in esophageal adenocarcinoma. *Carcinogenesis*. 2016 37:356-65;
- 3) Ceccarelli et al. Molecular Profiling Reveals Biologically Discrete Subsets and Pathways of Progression in Diffuse Glioma. 2016 *Cell* 164:550-563).
- 4) Cheng et al. (Integrative analysis of DNA methylation and gene expression reveals hepatocellular carcinoma-specific diagnostic biomarkers. *Genome Med*. 2018 10:42) have used a similar

approach selecting probes not methylated across a set of normal adjacent and other tumors that were methylated only in their cancer to select CpG sites as biomarkers.

This selection criterion has 3 issues: (1) due to field effect, “normal adjacent” pancreatic tissue may or may not be normal; (2) loci that are methylated in normal pancreas but become hypomethylated in tumors would be excluded from the analysis; (3) PanNETs presumably arise from the islets which comprise only 1-2% of pancreatic tissue. Therefore, the vast majority of the methylation signal detected from normal pancreata is from the acinar portion of the tissues as well as vascular tissue, stroma, and immune cells. Thus, the methylation status at a particular CpG locus could be very different in islet cells compared to the other cell types in the normal samples. For example, an unmethylated locus in islet cells would be excluded if the same locus was highly methylated in the exocrine portion of the pancreas.

Response:

- 1) We agree with the reviewer that the normal adjacent pancreata may not be normal. For the cases in this study as part of the International Cancer Genome Consortium (ICGC), these normal adjacent samples were tested on SNP arrays to identify any potential changes in allele frequency that could suggest alteration in copy number as a proxy for potential tumor contamination or early genomic changes that could indicate abnormal cells. They all passed this criteria. Also as we mention above, our approach was not to compare adjacent pancreata vs tumors. Our aim was to identify probes not methylated in other cells types in the surrounding pancreata. The probe selection would reduce confounding signal from non-tumour cells contaminating the tumour sample allowing the clustering of tumors and identification potential clinically relevant sub-groups. The sub-groups identified in our study have strong correlations with clinical and genomic features reassuring the potential of our approach.
- 2) The reviewer has a valid point, however if we were to use methylated sites in normal pancreata which have a reduction of methylation in tumors it would be much harder to identify loci with a tumor specific signal as tumor samples have some level of non-tumour cells. Tumor samples with different contributions of non-tumor cells might have different levels of methylation adding noise to the probes used for clustering.
- 3) The reviewer has another valid point. If our aim was to identify the role of methylation in the progression from islet cells to tumor to identify pathways of tumorigenesis, islet cells definitely would be the correct option. Instead, our aim was to evaluate if methylation could stratify tumors in a clinical setting. We believe that in this case, the use of islet cells and tumors to differentiate tumor sub-groups might result in confounding signals from contaminating non-tumor cells (acinar, vascular or immune cells). Our intent in this study was to identify tumor-specific methylation sites from clinical samples. Since our submission, a few papers have become available online that have addressed the question of cell of origin and potential progression pathways affected by methylation (Boons, et al. Cancers June 2020, and in biRxiv Domenico April 2020, Simon et al June 2020). Despite this not being our aim, we have evidence that the sub-groups identified may give an indication of cell of origin from RNASeq data (Revised Suppl. Fig 9), which are similar to results reported by Boons, et al. (2020).

We hope the above explanation clarifies the differences in the approach and in the research question between our study and recent publications.

To further reassure the reviewer of our approach, we also performed two new analyses using islet cells as controls.

Clustering of probes differentially methylated between islet cells and tumours cells

In the first analysis we performed clustering analysis using probes differentially methylated between 5 islet cells (3 beta and 2 alpha) from deceased donors (Neiman et al. PNAS 2017 114:13525-13530) versus our 84 tumors. Whereby tumor and islet cell were normalized together. A beta-mixture quantile (BMIQ)

normalisation was performed to account for probe type1 and type2 biases, followed by a quantile normalisation (QN). Further filtering was performed to remove probes in non-CpG sites, X or Y chromosome, SNP related polymorphisms and probes that map to multiple locations. After the filtering process, 411,159 probes were used for further analysis. A single value decomposition (SVD) analysis determined no significant batch effect. A total of 18,022 probes were differentially methylated between alpha/beta islet cells (n=5) and tumors (n=84). Significance was defined if the adjusted p-value <0.05 and the difference in the average beta value between the groups was >0.2. The methylation level of these 18,022 probes were dichotomized (representing a presence; beta value ≥ 0.3 , or absence: beta value <0.3 of methylation) and used to cluster the tumors. The unsupervised clustering revealed three groups of tumour samples (Figure a). The clinical and genomic features of these groups is shown (Figure b), and the relation of samples in these groups to genomic and clinical features originally associated with subgroups in the manuscript is shown for comparison (Figure c).

Figure a: Unsupervised clustering of tumors using dichotomized data for 18,022 probes differentially methylated between 5 islet cells (2 alpha and 3 beta) and 84 PanNET tumors (Cluster A in green, B in purple and C in orange). Colour bars indicate how tumors related to the original groups presented in the manuscript.

Figure b. Clinical features of the 3 sub-groups identified in Figure a. Tumor samples are presented in the same order of the clustering analysis (Figure a). Data is presented in the same fashion as the original manuscript for comparison. **No clear associations with genomic and clinical features.**

Figure c. Clinical and genomic features identified as associated with the groups in the original manuscript. Here the new groups identified by differential methylation between tumors and islet cells (Figure a) showed no relation with those clinical or genomic features.

In summary, the clusters identified here, using 18,022 differentially methylated probes between 5 islet cells (2 alpha and 3 beta) and tumors, are different from those identified in the manuscript and showed no association with the clinical or genomic features of the cohort (Figure c).

Clustering of probes differentially methylated between islet cells from two datasets and tumours cells

To try and increase the number of islet cells in our comparison we added data consisting of 87 islet cells from non-diabetic deceased donors (we assumed beta cells as their study evaluated the impact of DNA methylation in insulin secretion) (Hall et al. 2014 Genome Biology 15:522, GSE62640).

The cohort in this second analysis comprises:

- 1) Our original data (11 pancreata, 84 PanNETs),
- 2) Neiman et al. (2017) including 2 alpha and 3 beta cells, plus 1 acinar and 1 ductal cell.
- 3) Hall et al. (2014) 87 islet cells (53 males and 34 females with average age 56 and 58, respectively)

The total of 189 samples were normalized and filtered as described for the analysis above. Figure d presents a principal component analysis (PCA) using the entire set of 411,159 probes, which passed our filter and all samples in the cohort. We notice a high variability in the cohort of islet cells (Hall et al. 2014). A previous study reported 4 distinct subtypes of beta islet cell (Dorell et al. 2016 Nature communications 7:11756), which we assumed explained the variation observed. We then identified differentially methylated probes between 92 islet cells vs 84 PanNETs. A total of 12,323 probes were differentially methylated (adjusted p-value <0.05 and the difference in the average beta values >0.2).

The 12,323 differentially methylated probes were dichotomized and used to cluster all samples in the cohort (Figure e). The adjacent pancreata clustered with acinar cells as expected. We did identify 3 groups of islet cells, however, unexpectedly one group (I3) clustered closely with ductal cells. Islets cells in group I1 clustered with alpha and beta cells (Neiman et al. 2017), however closer to alpha than beta cells. This analysis led us to think that islet cells might not be pure. We then investigated the literature about the efficiency of islet cell isolation. Clardy, et al (2015 Sci Rep 5:13681) reported that beta cell isolation is generally complex, costly and/or does not clearly separate beta islet cells from other cells. Kim et al. (Transplantation Proceedings 2005, 37 3402-3403) reported the mean purity of islet cells was 54% across 69 cadaveric donors. The Hall et al. (2014) data used here reported that approximately 50% of the cells in their isolates were beta cells. These findings did not give us confidence to proceed with this analysis as islet cells methylation data might be a mixture of cell types.

Figure d. Principal component analysis of methylation levels across the entire genome for 189 samples (84 tumors, 11 adjacent pancreata, 87 islet cells, 3 beta and 2 alpha islet cells, 1 ductal and 1 acinar cell). As expected normal cells differ from tumors, but there is a high variability in the group of islet cells.

Figure e. Unsupervised clustering of dichotomized methylation of 12,323 probes differentially methylated between 92 islet cells and 84 tumors. Islet cells seem to separate in 3 clusters I1, I2 and I3. I1 seems to be

more similar to alpha cells than beta cells from (Neiman et al. 2017) and I3 more similar to Ductal cells. This could suggest a mixture of other non-islet cells. Acinar cells cluster closely to normal adjacent pancreata as expected.

In conclusion, there are very few methylation data sets available for islet cells, we have written to other authors but have received no response. In our opinion the corresponding adjacent pancreata will be more representative of the cells present in the surroundings of tumors and potentially more representative of non-tumor cells present in tumor samples.

We want to emphasise again that our aim was not to identify cell of origin of pathways of tumorigenesis but to identify tumor specific methylation in a clinical setting to allow tumor stratification. We hope the points made above and the analyses presented clarify our approach to the reviewer. We strongly believe that we took important steps 1) selecting probes from the normal pancreata that would have little or no contribution to the methylation signal within the tumor sample, 2) dichotomized the data before clustering to further remove potential contribution of non-tumor cells in the tumor sample. Together these approaches allowed for the identification of tumor sub-groups that have strong correlation with clinical and genomic features.

b. Probes were selected for analysis only if they mapped to the promoter region. Methylation may play a role in inactivation of distal enhancers or other regulatory elements (e.g. insulators) and should not be excluded from analysis for both subtype identification and correlation between methylation and gene expression.

Response:

We selected CpG sites in gene promoter regions due to the well accepted role of promoter methylation in the regulation of gene expression in a cell specific manner. Studies in several cancer types including PanNETs have shown hyper-methylation of promoter regions and global hypo-methylation in tumors. For the reasons mentioned above to try and minimize the contribution of signal of non-tumour cells we selected only promoter regions due to the probability of they show increased methylation in tumours. There are limitations to using regions (probes) with loss of methylation in the tumor as it is not possible to isolate the contribution from tumor cells from a mixed population of cells.

As enhancers can regulate gene expression independent of their location, distance or orientation (Banerji et al Cell. 1981; 27:299–308), we feel it is out of the scope of this manuscript to make any correlation between enhancers or other regulatory elements and gene expression. The correlation with gene expression was not limited to promoter probes. Probes used for correlation included vendor annotation for gene footprint, probes in promoter (TSS1500, TSS200, 5'UTR, First Exon), gene body or 3'UTR.

c. It would be worthwhile to rerun the analyses using a more comprehensive probe set.

Response:

Below we present the analyses of the data including more probes across the entire genome (as suggested by the reviewer), with following probe selection: 1) probes not methylated in adjacent normal pancreata across the entire genome, 2) from those the most variable across tumor ($SD > 0.20$) were used for clustering. A total of 7,227 most variable probes across 84 tumors and not methylated in the normal pancreata (independent of location in the genome) were dichotomized and used to cluster tumors (Figure f). Figure g shows the relationships between sub-groups and clinical and genomic features. Figure h shows association of the sub-groups with clinical and genomic features. We hope this confirms our original approach as similar groups are identified. Note that this analysis includes the 3,378 probes used in the original manuscript plus 3,849 probes in other locations in the genome, thus some similarity is expected. Here 15 samples (17.9%) moved clusters compared to our original analysis (Figure f). Please note that the

order of the clusters are different when comparing with the original manuscript (Original data: T1-wild type/harbour most insulinomas **now C**; T2- ATRX/DAXX/MEN1 recurrent LOH of half of the genome, ALT and worse prognostic markers **now A**, T3- MEN1 recurrent LOH chr11, better prognostic markers **now B**). Here 11 tumors originally clustered in ATRX/DAXX/MEN1 recurrent LOH of half of the genome with worse prognostic markers and 3 from MEN1 recurrent LOH chr11 groups now cluster with the wild type/insulinoma group (T3 orange, Figure f). The samples that changed groups form a distinct sub-group in group C (Figures f, g) with mutations, LOH, longer telomeres and ALT. None of the samples in the wild type group from the original analysis changed groups in this new reanalysis. Clinical associations are similar to our original analysis (Figure h). Our take in this case would be to use a lower number of sites for clustering (**original analysis**), if our findings could be confirmed in an independent cohort the next and ideal step is to identify a much smaller informative set of CpG for clinical use. Increasing the number of sites for this analysis may not be useful in the future in a clinical setting. We hope that with the clarification about our approach to probe selection and the extra analyses presented in this rebuttal we can reassure the reviewer that the approaches used in our original manuscript are sound and valid.

We have added the analysis below as Supplementary Figure 2.

Page 4, Line 159: Similar subgroups were identified when selecting probes across the entire genome (Supplementary Figure 2)

Figure f. Unsupervised clustering of dichotomized methylation of 7,227 probes. Probes were selected if not methylated in 11 normal adjacent pancreata (<0.30 , independent of genomic location) and most variable probes across 84 tumors ($SD > 0.20$) across the entire genome. Three groups were identified A (green), B (purple) and C (orange). Black underscore lines show samples that changed subgroups from the original analysis presented in the manuscript. Colours in the bars show the original groups presented in the manuscript.

Figure g: Clinical features of the 3 sub-groups identified in Figure f. Tumors are presented in the same order of the clustering analysis (Figure f). Data is presented in the same fashion as the original manuscript for comparison. Black underscore lines show samples that changed subgroups from the original analysis presented in the manuscript.

Figure h. Clinical features with significant association in the original manuscript for comparison. We can see similar trends here. Please consider that the groupings tested here were obtained from a clustering analysis that included the original 3,378 most variable promoter probes used in the original data. We would expect to see some similarity.

2. It appears that a significantly larger set of probes was used for the identification of differentially methylated loci (DMLs) between subtypes. DMLs were then discussed in relation to the 3 subtypes identified by a smaller set of probes. This seems odd to this reviewer and the rationale for doing this should be made clear in the Results section.

Response:

The use of a smaller number of most variable probes across tumors to identify groups, and once an association with clinical features is identified, then proceed with differential methylation between groups for the entire genome, is not a novel approach. Please see two examples of more recent publications:

- Gao et al. (“Genome-wide DNA methylome analysis reveals methylation subtypes with different clinical outcomes for acute myeloid leukemia patients”, Cancer Medicine 2020) used the top 10% most variable probes across tumors then used a differential methylation analysis for the entire genome between the groups.
- Simon et al (bioRxiv, June12, 2020. “An Integrative Genetic, Epigenetic and Proteomic Characterization of Pancreatic Neuroendocrine Neoplasms (PanNENs) defines Distinct Molecular Features of α - and β -cell like Subgroups”) used 8,000 most variable probes across tumors to identify subtypes and then compared the subtypes for more than 800,000 positions (EPIC arrays) to identify differentially methylated probes (DMP) between subtypes.

We have added sentences to our results and methods sections to clarify the rationale of our approach.

Page 4, Line 151: “To identify cancer specific DNA methylation that could potentially stratify PanNETs, we selected CpG sites located in promoter regions which were not methylated (beta value <0.3) across all 11 normal adjacent pancreata samples.”

Page 5, Line 200: “Next we compared the 3 subgroups (identified by 3,378 most variable probes across tumors and not methylated in normal adjacent pancreata) to identify differentially methylated sites across the entire genome and gain some potential insight into the biology that may differentiate the groups.”

Page 12, Line 424: “The approaches described above were taken to minimize potential confounding signal from non-tumor cells in the tumor samples.”

Page 12, Line 429: “Differential methylation analysis between identified sub-groups aimed to gain biological insights about the differences in those groups.”

Minor points:

1. Several terms in the text and/or in the Figure legends should be clarified. For example, the less informed reader shouldn't have to look up the meaning and/or significance of “functional” tumors, ALT, telomere ratio ..

Response:

These definitions and information were added to the label and legend of Figure 1.

Text added: “ ALT status: alternative lengthening of telomeres assessed using C-Tailing qPCR²; ALT+ve: positive for ALT, ALT-ve: Negative for ALT; Telomere Ratio: Reads with telomeric repeats were counted in both the tumor and matched normal sample and normalized to the mean genomic coverage of the sample using qMotif² for both the tumor and matched normal sample and the ratio gives us an indication of shortening or lengthening in relation to normal sample. Functional PanNETs: tumors that overproduce biologically active hormone.”

2. In line 184, clarify the meaning of “if the observed methylation profile is driven by copy number,”. And in line 189, what does “hypo-methylation” refer to. Hypo-methylation is a relative term. Do the authors mean to say that apparent hypo-methylation may result from copy number loss?

Response:

The text on page 5 was altered.

Line 189: “We next investigated if clustering using tumor methylation could be influenced by the different profile of copy number observed between the groups, more precisely if loss of copy number would result in loss of methylation.”

Line 195: “We did not observe lower levels of methylation of CpG sites in tumors in the T2 subgroup, which harboured most tumors with recurrent loss or LOH across 11 chromosomes compared to the levels of methylation in the other subgroups (T1 and T3).”

3. Line 211, what is meant by “the footprint of ”. Do you mean, the gene body alone, or does it include the promoter regions?

Response:

Extra information has been included in the paper about the region we called gene footprint.

Page 6 Line 218: “From the 11,042 GpG sites differentially methylated between at least one pair of subgroups, 7,303 CpG sites mapped to the footprint of 4,317 genes (regions TSS1500, TSS200, 5’UTR, First Exon, gene body or 3’UTR).

4. In Fig. 4B, the methylation level is more appropriate on the x-axis; same for Suppl. Fig. 6B&C

Response:

Graphs were replotted as suggested by the reviewer and figures updated. Please note due to changes made in the manuscript this is now Figure 3b and Supplementary Figure 7b and c.

5. Fig. 4A may be more suitable as a supplemental fig.

Response:

Please note due to changes made in the manuscript this is now Figure 3a.

We would prefer to keep this panel as a main figure. *MGMT* expression is highly relevant in the clinical setting as a potential marker for treatment. Our results are the first to show that methylation impact on expression might not be from the methylation status of promoter but the loss of body methylation and LOH which correlates positively with its expression. Figure 3a shows clearly the patterns of methylation and copy number across the tumors. We believe this has strong clinical relevance considering that there is a clinical trial evaluating if *MGMT* promoter methylation could predict response to alkylating agents in well-differentiated duodenum, pancreatic, lung, or unknown primitive NETs (NCT03217097). Our initial findings need to be validated with other datasets, as this might influence how to select patients for treatment.

6. Line 229-230, Please clarify what is meant by “Low methylation in the gene body together with LOH could potentially drive low expression of MGMT within the T2 subgroup”. It’s not clear if low methylation and LOH occur at the same time in a tumor, or independently.

Response:

Methylation and genomic data was obtained from the same DNA extraction for the tumour samples. The tumors in the T2 sub-group present higher frequency of loss or LOH of 11 chromosomes including *MGMT* loci (Revised_Figure 3a) and lower levels of methylation in probes located in the gene body of *MGMT* compared to the other subgroups. The order that these events occur during tumor development is out of the scope of this manuscript. Here we can only say they are both present in the majority of T2 tumors at the sampling time (i.e. when the tissue was collected). In Supplementary Figure 5 we show that the pattern of methylation of the two genes flanking *MGMT* gene do not present the same pattern across the subgroups suggesting that the loss of methylation on *MGMT* body is not a result of the loss of one allele but suggestive of a coordinated joint mechanism of LOH and loss of methylation in the body of the gene that results in lower MGMT expression (Fig 3b, 3c). These results suggest that these tumors might respond to Temozolomide. We have edited the text to make the point clear.

Page6, Line 232: Gene body methylation of *MGMT* was significantly lower in tumors in the subgroup T2 compared to tumors in the other groups (T1 and T3), T2 tumors also harboured recurrent loss of chromosome 10q arm including the loci of MGMT (Figure 3a).

Page10 Line 368: Methylation data was generated using the same DNA extraction that was used for the whole genome landscape of these tumors that has previously been characterized by Scarpa et al.². Their study included a description of the tumor content, copy number profiles, somatic and germline mutations, together with clinical parameters.

Reviewer #2 (Remarks to the Author):

Lakis et al. report DNA methylation analysis of a large cohort of sporadic PanNETs. The cohort is impressive, the analysis is thorough and very informative and the results are important. The authors are well known for their expertise in PanNET genetics. However, our ability to make robust conclusions from this analysis are limited, mainly due to the first comment, but also in the absence of proper referencing for other recent publications in the field. Here are few points to be addressed in the view of the reviewer:

Major

Adjacent pancreata cannot be used as control for DNA methylation analysis of islet cell tumors, as the former include mainly exocrine pancreas tissue. This selection is very surprising and should be substituted by normal islet cells samples (derived from cadavers, which are also not the perfect choice but much better than practically a different organ). DNA methylation is tissue (or even cell-) specific. Since the entire analysis is based on these tissue samples, the results might be entirely flawed and the analysis should be probably re-done from the square 0.

Response:

We appreciate the reviewer’s concerns, but our aim here was not to identify differences between tumor and normal pancreata. By identifying probes not methylated in normal adjacent pancreata and then selecting from those the most variable probes across tumors our aim was identify probes that would suffer little or no influence from non-tumor cells (surrounding pancreata). This approach aims to identify tumor specific methylation to stratify tumours in a clinical setting, where tumor samples might contain

some level of non-tumor cells. The reviewer has a valid point that cells have specific methylation, so our approach was to identify probes that are not methylated across all normal adjacent pancreata samples (with a variety of cell types). Please refer to the response to reviewer 1 and extra analyses presented above. Here we did not use normal to identify cell of origin or tumor progression pathways. This has been addressed in recent reports that came out in a similar time frame as our submission (Boons, et al. Cancers June 2020, and in biRxiv Di Domenico April 2020, Simon et al June 2020). Here our aim was to identify CpG sites that could differentiate tumor types in the clinical setting where tumor samples have some level of non-tumour cells that could confound tumor signal.

How much of the data presented and analyzed derived from the data reported in Scarpa et al. Nature 2017?

Response:

Methylation data of 84 tumor and 11 normal Pancreata and RNASeq (gene expression) of 20 out the 47 PanNETs were not reported in Scarpa et al (Nature 2017). We have reported in the text that all clinical and genomic data was from Scarpa et al. Please see below.

Page 4 line 143: “The clinical and genomic information from Scarpa et al.² (Supplementary Table 1) was used to gain insights into potential PanNET subgroups identified by methylation patterns.”

Page 5 Line 183: To investigate how copy number alterations relate to the methylation subgroups (T1-T3), we compared copy number data generated by Scarpa et al.² with the methylation data (Figure 2).

Methods:

Page 10 line 368: “Methylation data was generated using the same DNA extraction that was used for the whole genome landscape of these tumors previously characterized by Scarpa et al.². Their study included the description of the tumor content, copy number profiles, somatic and germline mutations, together with clinical parameters. Here we used this publicly available information to complement analysis of methylation patterns across tumors.”

Page 12 Line 441: RNASeq data were available for a subset of 47 cases in the present cohort. RNASeq data of 27 cases were previously published by Scarpa et al.² and a new extra set of 20 cases formed the set of 47 cases with gene expression data under (<https://www.ebi.ac.uk/ega/search/site/EGAD00001006063>).

It is important to assess/validate/reject the data recently reported by Boons, Vandamme et al regarding Pdx1 and Arx promoter methylation and link with tumor aggressivity/prognosis (Cancers, 2020).

Response: Boons et al (June 2020) compared tumors with 5 islet cells (2 alpha and 3 beta, from Neiman et al 2017). In their pathway analysis of differentially methylated genes, there was a strong signal of immune related pathways that may suggest signal from non-tumor cells in their data set. Based on our analysis above and reports in the literature about isolation of islet cells, these findings are not surprising. They only evaluated *PDX1* in their manuscript, using probes located in this gene to cluster tumor samples.

Boons et al (2020) did not evaluate *ARX* gene. *ARX* gene is located in chromosome X, and due to the role of methylation in chromosome X inactivation in females, it is common practice to remove chromosomes X and Y from methylation analyses with mixed cohorts. To our knowledge no study has shown *ARX* promoter methylation. Chan et al (2018) have shown a difference in gene expression of the *ARX* gene. Cejas et al. (2019) has evaluated *ARX* expression by immunohistochemistry. We did not observe *ARX*

differential methylation. The new analyses of *ARX* gene are presented in Supplementary Figure 8 and Supplementary Table 7 added to the revised manuscript.

We have added a reference to Boons et al. comparing their results with ours.

Page 9 Line 332: “Boons et al¹⁵ have used methylation probes in the *PDX1* loci to identify two PanNET subtypes (A and B), which they reported similar to alpha and beta cell islet cells, respectively. Subtype A had a significant worse prognosis compared to subtype B, with most insulinomas in the latter group. Subtype B was mostly wild type for *ATRX/DAXX/MEN1* genes, which is in agreement with our T1 subgroup. In our study we further identified two groups with different clinical and genomic features. Gene expression and methylation patterns of these groups (T2 and T3) suggests potential similarity with alpha islet cells. In Boons et al¹⁵ subtype A seems to have a further branching in their cluster analysis that suggests a potential difference in metastatic disease, but the authors did not further discuss that. Here we speculate that further subgrouping in their subtype A could be related to the two groups (T2 and T3) seen in our study. T2 and T3 are mostly non-functional PanNETs but showed significant differences in histological parameters indicative of prognosis, with T2 associated with markers of worse prognosis.”

Page 7, Line 256: As *ARX* gene is located in the X chromosome, which is normally removed from methylation analysis, we evaluated *ARX* methylation patterns separately. We did not observe differences in the methylation levels between the groups (Supplementary Figure 8) even when evaluating males and females separately (Supplementary Table 7). However, we observed a higher gene expression in T2 and T3 compared to T1 (wild type tumor). This is in agreement with previous studies^{4,22}, and suggests that the methylation status of *ARX* may not affect expression of the gene.

Also, please refer to the recent data uploaded to BioRxiv by Di Domenico and co on PanNET methylation analysis.

Response:

We have added the following references to support the new Supplementary Figure 10, Boons, et al. Cancers June 2020, and in biRxiv Di Domenico April 2020, Simon et al June 2020

Page 7, Line 270: The identification of the cell of origin was not the primary aim of the present study, but has been integral to recent studies¹⁵⁻¹⁷ that have identified similarities of PanNET tumors with alpha and beta islet cells and pathways of tumorigenesis. With this in mind we compared 5 islet cells (2 alpha and 3 beta islet cells)²⁴ with our subgroups. Supplementary Figure 10 shows that methylation of a set of probes with variable methylation between alpha and beta islet cells suggest T1 tumors to be beta like tumors and T2 and T3 tumors to be alpha like tumors in agreement with gene expression profiles in the subset of 47 tumors with RNASeq and recent publications.

Page 3 Line 130: More recently, studies have suggested that methylation can identify cell of origin for PanNETs¹⁵⁻¹⁷.

lines 217-220 – what does DNAm and gene exp correlation (positive or negative?) reflect? This is a known mechanism of transcription regulation. Is an association with clinical parameters.

Response:

We have added extra data on Supplementary Table 6 with gene expression and methylation levels in each subgroups to exemplify the relationship with subgroups identified and their clinical features.

Page 6, Line 226: Correlations were positive and negative depending on the gene and location of differentially methylated probes in the gene. For details of gene expression and methylation in each subgroup see Supplementary Table 6.

Line 221 – lower compared to which group(s)?

Response:

We have add text to clarify.

Page 6, Line 232: Gene body methylation of MGMT was significantly lower in tumors in the subgroup T2 compared to tumors in the other groups (T1 and T3), T2 tumors also harboured recurrent loss of chromosome 10q arm including the loci of MGMT (Figure 3a).

Line 224-225 – In line 224 the authors wrote “positive” and probably meant “negative? how do the authors explain positive correlation between promoter methylation and gene expression?

Response:

Positive is correct. Most differentially methylated probes in the *MGMT* gene are located in the body of the gene. Normally promoter probes show negative correlation and body probes show positive correlation with gene expression. We did see only one probe in the promoter differentially methylated with a positive correlation. This is highlighted in Page 6, Line 236: “Only one CpG site (cg24420981) located in the promoter region of MGMT was differentially methylated between at least one pair of subgroups, and unexpectedly showed a positive correlation with expression.”

We believe this positive correlation is only a consequence of the body methylation and LOH driving expression of the gene.

Lines 246-247 – Removing the x chromosome is a reversible action. I strongly suggest to include this analysis.

Response:

We respectfully disagree with the reviewer’s comment to include probes on X-chromosome in our analysis. In our study we have 32 female and 52 males. DNA methylation is a known key factor in X chromosome inactivation in female cells to compensate for the extra X chromosome compared to male cells. There are studies showing differences in the levels of methylation of X-chromosome in different tissues and cell types. Hall et al (Genome Biology 2014, 15:522) showed chromosome-wide DNA methylation level of the X-chromosome is higher in female versus male islet cells. It is common practice in the analysis of DNA methylation when evaluating a mixed cohort of females and males to remove probes on X and Y chromosomes. See two recent examples Capper et al 2018 Nature 555:469-474 and Simon et al bioRxiv (June12, 2020). An Integrative Genetic, Epigenetic and Proteomic Characterization of Pancreatic Neuroendocrine Neoplasms (PanNENs) defines Distinct Molecular Features of α - and β -cell like Subgroups.

Page 7 Line 256: As ARX gene is located in the X chromosome, which is normally removed from methylation analysis, we evaluated ARX methylation patterns separately. We did no observed differences in the methylation levels between the groups (Supplementary Figure 8) even when evaluating males and females separately (Supplementary Table7). However, we observed a higher gene expression in T2 and T3

compared to T1 (wild type tumour). This is in agreement with previous studies^{4,22}, and suggests that the methylation status of ARX may not affect expression of the gene.

Minor

60% survival in 5 years: Maybe only for advanced disease, or old data when the PanNET incidentalomas were less frequent. Please add data to support other than ref 1.

Response:

We changed our reference and text. Now reads:

Page 3, Line 110: The overall 5 year survival rate for PanNET patients is 54%, but varies greatly according to stage at diagnosis. Patients with local disease have the 5 year survival rate is 93%, while patients with locally advanced disease is 77%, and for those with distant metastatic disease is 27% (www.cancer.net/cancer-types/neuroendocrine-tumor-pancreas/statistics; <https://www.cancer.org/cancer/pancreatic-neuroendocrine-tumor/detection-diagnosis-staging/survival-rates.html>).

Lines 290-291 higher proportion than what?

Response:

Extra information was added to the sentence.

Page 9 Line 310: "T3 tumors had a lower incidence of perineural, vascular invasion and extra-pancreatic spread, and a higher proportion of grade G1 tumors compared to the other groups (T1 and T2), suggesting these tumors have less aggressive behaviour¹⁰."

The figures are of low quality. There are no dendrograms in the heat-maps, was this unsupervised clustering?

Response: Original clustering of the tumor samples was performed on dichotomized data as presented in Supplementary Figure 1. All the other Figures and heatmaps, tumors (columns) are plotted in the same sample order of the clustering presented in Supplementary Figure 1.

We have added extra info in the Figures legend:

Figure 1: Text added: Tumor (columns) are presented in the same order as dichotomized clustering presented in Supplementary Figure 1.

Figure 2: Text added: Tumor (columns) are presented in the same order as dichotomized clustering presented in Supplementary Figure 1.

Figure 3: Tumor (columns) are presented in the same order as dichotomized clustering presented in Supplementary Figure 1. Probes are plotted by genomic coordinates from 5' to 3' direction.

Please re-consider whether all figures are contributory and informative.

Response:

We have removed original Figure 3 showing differentially methylated probes between subgroups.

Reviewer #3 (Remarks to the Author):

Lakis et al report in an international collaboration the correlation between methylation profiles and clinical / genetic data of a large number of PanNETS. They were able to identify three subgroups with distinct patterns of methylation.

They extensively investigated the relation of methylation with commonly mutated genes in these tumors, somatic copy numbers changes and gene expression

In this, to date largest cohort, they were able to show new insights in both genetic and epigenetic landscape of panNETS.

I have a few questions/ suggestions:

1) with this strong distinction between the three groups, the authors might give a stronger recommendation show to treat these patients in clinical practice

Response:

We have added to the discussion section what we think are the next steps to confirm methylation as a potential biomarker for PanNETs for potential future use in the clinic.

Page 9, Line 321: Our findings are the initial report of the potential for DNA methylation as a biomarker in PanNET, which needs to be validated in other cohorts. If validated, the next step towards the use in clinical practice is the identification of a smaller number of CpG sites with PanNET specific methylation that can specifically distinguish the subtypes that could in the future influence how the patients are managed in the clinic for treatment and surveillance.

2) the cohort only consist of sporadic tumors. Do the authors feel that their data can be extrapolated to familial PanNETs, e.g. MEN 1 related tumors? Please discuss.

Response:

Tirosh et al. (2019 Cancer 125:1247-1257) evaluated methylation of 9 sporadic *MEN1* tumors, 10 *MEN1* and 10 *VHL* hereditary PanNETs. They reported that sporadic and hereditary *MEN1* are more similar than *VHL* tumors. However, they had two groups of *MEN1* mutant tumors, one enriched for sporadic tumors and the other for hereditary *MEN1* tumors. Our cohort had only 4 *MEN1* germline tumors. Two of those clustered with somatic mutant *MEN1* tumors (T3) and two with the T2 (*ATRX/DAXX/MEN1*) mutants. Due to the low number of germline mutations in our cohort and the variability of clinical features of familial cases, we think it is too early to make any extrapolation to familial cases. This would be a really interesting question for future studies.

Page 9, Line 312: *MEN1* mutations also play a role in inherited PanNETs. A previous study⁷ evaluating methylation of 9 sporadic and 10 inherited *MEN1* related PanNETs suggested that *MEN1* mutated tumours in both settings are more similar than *VHL* inherited tumours (n=10). But sporadic and inherited *MEN1* related PanNETs have distinct patterns of methylation. It would be interesting in the near future to evaluate how *MEN1* related inherited PanNETs compare to the two subgroups, harbouring *MEN1* mutations identified here, which have distinct histological parameters indicative of prognosis.

3) in the introduction section, in both line 114 and 119, they use the word conversely. They might consider to change one of them.

Response:

Text was edited.

Page 3, Line 118 now reads: “Non-functional PanNETs, due to the lack of early symptoms, are normally detected at later stages with locally advanced or metastatic disease. Non-functional PanNETs are a more heterogeneous group of tumors with unpredictable and varying degrees of malignancy^{1,10}. The best predictor of clinical outcome of PanNETs is the fraction of proliferating neoplastic cells, with high grade (G3) tumors being more aggressive disease and with molecular alterations that often align them with neuroendocrine carcinomas¹. Conversely, low (G1) and intermediate (G2) grade PanNETs have a distinct molecular pathology and lack reliable biomarkers to assist prediction of malignancy and selection of treatment¹¹.”

4) discussion section line 297: typo ATRX instead of ATX.

Response:

Typo was corrected.

Reviewers' Comments:

Reviewer #1:

Remarks to the Author:

Lakis et al. 2020, revised version

In a very comprehensive rebuttal letter, the authors describe extensive analyses and explanations for their experimental approach and provide compelling arguments for their choice of CpG loci included in the study and the use of normal pancreas as a control. Justification for this approach is made much clearer in the revised manuscript.

A few minor points:

1. Line 203, "compare" should be "compared"
2. Line 239, "mapped to MGMT gene " should be "mapped to the MGMT gene body"
3. Line 240, "positive" should be "positively"
4. Line 243, "unexpectedly showed" may be better as "unexpectedly also showed"
5. Line 263, "no" should be "not"
6. Line 277, "identify" should be "identified"?
7. Line 352, speculate how gene body methylation of MGMT may affect expression?

Reviewer #2:

Remarks to the Author:

The authors have addressed some of the comments. However, the main concern of this review is usage of normal pancreata as reference.

The authors suggest that they wanted to identify specific hypo/hypermethylated probes in tumors. However, by such comparison, any gene that is differentially methylated between islet cell and normal pancreas (and again, vs. any other tissue) will pop-out as significant. Relying on the representation of islet cells in the reference data is not sufficient considering their very little fraction in pancreas tissue. This was also independently and very clearly stated by reviewer 1.

I suggest, again, to consider using a valid reference for this analysis, as otherwise its interpretation may be flawed and even misleading.

Reviewer #3:

Remarks to the Author:

Colleague Nones et al have carefully addressed all questions raised and in this form the manuscript is suitable for publication.

Please find below our responses (blue font) to reviewer's comments (black font) about our manuscript "DNA methylation patterns identify subgroups of pancreatic neuroendocrine tumors with clinical association"

Thank you for considering our revised manuscript for publication in Communications Biology. We hope the responses and changes made to the revised version would address the last few questions from reviewers.

Reviewer #1 (Remarks to the Author):

Lakis et al. 2020, revised version

In a very comprehensive rebuttal letter, the authors describe extensive analyses and explanations for their experimental approach and provide compelling arguments for their choice of CpG loci included in the study and the use of normal pancreas as a control. Justification for this approach is made much clearer in the revised manuscript.

A few minor points:

1. Line 203, ???compare??? should be ???compared???
2. Line 239, ???mapped to MGMT gene ??? should be ???mapped to the MGMT gene body???
3. Line 240, ???positive??? should be ???positively???
4. Line 243, ???unexpectedly showed??? may be better as ???unexpectedly also showed???
5. Line 263, ???no??? should be ???not???
6. Line 277, ???identify??? should be ???identified???

Response: We incorporated into the manuscript all suggestions made above by the reviewer.

7. Line 352, speculate how gene body methylation of MGMT may affect expression?

Response: We have added text and references about the potential mechanisms of how gene body methylation might affect gene expression.

Page9 line 317: "The role of methylation in the body of genes and its relationship with gene expression is not fully understood. Studies have suggested potential mechanisms such as regulation of alternative promoters, regulation of retrotransposon elements influencing alternative transcription and other functional elements that maintain efficiency of transcription³¹⁻³³. *MGMT* gene body methylation would require further investigation due to its potential clinical impact as a predictive marker for treatment of PanNETs."

Reviewer #2 (Remarks to the Author):

The authors have addressed some of the comments. However, the main concern of this review is usage of normal pancreata as reference.

The authors suggest that they wanted to identify specific hypo/hypermethylated probes in tumors. However, by such comparison, any gene that is differentially methylated between islet cell and normal pancreas (and again, vs. any other tissue) will pop-out as significant. Relying on the representation of islet cells in the reference data is not sufficient considering

their very little fraction in pancreas tissue. This was also independently and very clearly stated by reviewer

1. I suggest, again, to consider using a valid reference for this analysis, as otherwise its interpretation may be flawed and even misleading.

Response: We want to assure the reviewer that we did neither perform differentially methylation analysis between normal adjacent pancreata and tumors to select probes for tumor clustering nor did we use the normal pancreata as a reference.

Our intent was to identify probes not methylated across 11 adjacent pancreata and select from those the probes with most variable levels of methylation across tumors. This probe selection was to minimize confounding signal from normal contamination (mostly acinar and ductal cells) in the tumor sample. Most tumor samples have a variable level of non-tumor cells, this is not particular to our cohort but observed in most cancer studies. These most variable probes across tumors were used to identify potential groups and these groups were correlated with clinical and genomic features. The Figure below shows the levels of methylation for the most variable probes (n=3,378) across tumors, normal adjacent and different cell types (including islet cells). In this set of probes, some probes have higher methylations in islet cells and tumors than in the normal pancreata. This suggests that tumors have a more similar pattern of methylation to islet cells, as expected. This reinforces that our probe selection does not influence tumour biology only aims to remove any potential confounding methylation signal from the content of non-tumor cells.

We added a sentence to the results section Page3 line 116: “With the probe selection described above we expected to reduce potential confounding signal from non-tumour cells present in the tumour samples.”

Figure 1. Methylation levels of 3,378 probes used to identify tumour subgroups. Tumors are presented in the order of the cluster (Supplementary Fig.1). Probes (rows) were clustered in the tumors using Ward's clustering method. Levels of methylation in normal adjacent pancreata and other cell types (Neiman, et al. 2017) are plotted on the right.

We agree with the reviewer that the use of methylation in the future as a potential biomarker to stratify patients requires further validation in other PanNET cohorts. There is also the need for a further reduction of the most informative CpGs by identifying sites uniquely methylated in PanNET and not in other tissues. However, our study is the first to show the presence of methylation subgroups. This was an exploratory overview of a well-annotated cohort with genomic and clinical data. There is a long process to validate and improve the use of methylation as a potential biomarker in PanNET. We had a sentence in the discussion highlighting this and we have now added text (in red) in the revised version:

Page 8 line 282: "Overall, the results presented here advance the comprehension of the genetic and epigenetic landscape of PanNETs, indicating also that patterns of methylation have the potential to stratify PanNETs prognosis. Our findings are the initial report of the potential for DNA methylation as a biomarker in PanNET, which needs to be validated in other cohorts. If validated, the next step towards the use in clinical practice is the identification of a smaller number of CpG sites with PanNET specific methylation **compared to other tissues** that can specifically distinguish the subtypes and could in the future influence how the patients are managed in the clinic for treatment and surveillance."

Reviewer #3 (Remarks to the Author):

Colleague Nones et al have carefully addressed all questions raised and in this form the manuscript is suitable for publication.

Response: This reviewer requested no further changes.